# Predicting mean depth and area fraction of Antarctic supraglacial melt lakes with physics-based parameterizations

Danielle Grau [1] ✉, Azeez Hussain[2,3] & Alexander A. Robel[1,3]

Despite the importance of supraglacial melt lakes to the future evolution of polar ice sheets, they are not represented in current large-scale climate and ice sheet models. In this study, we use ICESat-2 satellite surface elevation measurements to show that roughness on the Antarctic Ice Sheet surface is largely self-affine. Estimation of ice sheet surface roughness statistics then enables the development of a set of simple mathematical expressions parameterizing the average supraglacial melt lake area fraction and lake depth from statistical fitting of large simulation ensembles of water flow on random, self-affine surfaces. These parameterizations provide predictions that are generally consistent with observations, with some exceptions. Finally, we predict that on large portions of Antarctic ice shelves supraglacial lakes are likely to, on average, stay less than one meter deep and occupy less than 40% of the ice area, absent changes in ice shelf surface roughness.

Over the past three decades, observations have shown widespread surface melting and supraglacial lake formation on Antarctic ice shelves[1], driven by atmospheric warming[2]. Atmospheric temperatures over Antarctica are expected to continue increasing in the future, along with an increasing prevalence of melt lakes on the ice sheet surface[3–6]. Supraglacial melt lakes have been implicated in the thinning and collapse of floating ice shelves in Antarctica, with the prominent example of the Larsen B ice shelf collapse during the 2001–2002 melt season[7–9]. In the days before the collapse, many supraglacial melt lakes formed on the ice shelf surface, which is hypothesized to have caused hydrofracturing across the ice shelf and its eventual disintegration[7,9,10].

Ice sheet surface melt affects the surface energy balance of glacial surfaces. Darkening of ice by water (i.e., decreasing surface albedo) causes increased absorption of incoming shortwave radiation by the ice and snow. Ice sheet surface melting initiates a positive feedback, whereby a lower surface albedo allows for more shortwave absorption, further enhancing meltwater production and deepening lakes. After refreezing, albedo is permanently altered from its original value due to snowpack and firn density changes from meltwater saturation, which promotes future melt lake formation at these locations[11]. Additionally,

supraglacial melt lakes alter the latent and sensible heat fluxes through air-ice sheet interactions[12,13].

Meltwater from supraglacial lakes may also fill surface fractures in the ice sheet, causing them to propagate deeper[9,14–16]. Incorporating supraglacial melt lakes into large-scale models is thus vital in properly simulating the potential interaction between surface melt and ice sheet fracturing and calving. Supraglacial melt lakes are currently omitted from large-scale models because they are typically small ($10^1$–$10^3$ meters across)[17] compared to the grid spacing in climate ($10^4$–$10^5$ meters) and ice sheet models ($10^3$–$10^4$ meters). Existing numerical models of supraglacial hydrology are computationally expensive to run over entire ice sheets at decadal or longer time scales[11,18,19]. Furthermore, no physically-based theories or empirical parameterizations exist that capture the area-averaged effects of supraglacial melt lakes on the surface energy balance and ice sheet fracturing. This study aims to remedy these gaps through the development of supraglacial melt lake parameterizations that can be implemented in large-scale models.

In this study, we develop simple equations parameterizing the spatially averaged size of supraglacial melt lakes, which can be added to large-scale models at negligible computational expense. The core idea behind these parameterizations derives from prior theoretical

[1]School of Earth and Atmospheric Sciences, Georgia Institute of Technology, 311 Ferst Dr., Atlanta, GA, USA. [2]School of Physics, Georgia Institute of Technology, 837 State St. NW, Atlanta, GA, USA. [3]These authors contributed equally: Azeez Hussain, Alexander A. Robel. ✉e-mail: dgrau7@gatech.edu

advances from percolation physics[20,21] and terrestrial hydrology studies which find robust statistical relationships between lake size and parameters describing surface roughness (on land or otherwise)[22]. In this work, we define this surface roughness through the self-affinity of the glacier surface. Self-affinity is a fractal property exhibited by most of the Earth's surface[22,23]. This property, quantified by the Hurst exponent, measures how the vertical scale of topography varies across horizontal scales[23].

We start by using satellite altimetry measurements to show that surface roughness is self-affine over the vast majority of the Antarctic Ice Sheet. We quantify relevant roughness parameters for the grounded and floating portions of the ice sheet. We then develop a set of simple mathematical parameterizations through a large Monte Carlo ensemble of physics-based simulations of supraglacial meltwater flow over self-affine surfaces. We validate these parameterizations by comparing melt lake area fraction and depth predictions to observations from satellite imagery on two Antarctic ice shelves. Finally, we estimate the maximum attainable mean melt lake depth and area fraction for the current roughness characteristics of the Antarctic Ice Sheet. The methods used throughout this study are described in more detail in the "Materials and Methods" section. In short, this study confirms that ice sheet surfaces, like other geomorphic surfaces, are self-affine and provides a set of supraglacial melt size parameterizations that can be implemented in large-scale models.

## Results

### Self-affinity of Antarctic surface roughness from ICESat-2

Self-affinity describes a repeating pattern within an object, sequence, or surface that occurs over a wide range of scales (also referred to as "fractal" or "self-similar") (Fig. 1). Analysis of land surface elevation measurements shows Earth's surface roughness is largely self-affine[23,24], extending even to the land surface beneath ice sheets[25]. The self-affinity of Earth's land surface has been used to develop elegant predictions for the size distribution of terrestrial lakes filling

depressions on this surface[22,24,26]. In this section, we investigate whether the Antarctic ice sheet surface also exhibits such self-affinity, and use satellite altimetry measurements to quantify two statistical roughness properties of the ice sheet surface: Hurst exponent ($H$) and standard deviation ($\sigma$).

We estimate surface roughness properties across the Antarctic ice sheet, using the entire Antarctic ICESat-2 ATL06 land-ice elevation data catalog from June 2021 to June 2022 (described further in "Materials and Methods"). This one-year period includes four repeats for each track[27], and we found little temporal variation in surface roughness parameters over that period. All sub-tracks with calculated Hurst exponents less than zero or greater than one are discarded, since they cannot be reliably described as self-affine (i.e., outside the acceptable confines of the range for the Hurst exponent). Typically, such cases are related to spurious elevation data causing large deviations in the power spectral density. Additionally, any linear regressions with $R^2$ values less than 0.7 are discarded to ensure that the only sub-tracks considered are those in which the data strongly indicate roughness is self-affine. Of the 1.8 million sub-tracks that were analyzed, 7.3% of the sub-tracks were discarded during the analysis process. 7.1% were discarded due to poor $R^2$ values when fitting, and the remaining 0.2% were discarded due to the calculated Hurst exponents being outside the acceptable range of [0,1]. As a result, over 92% of all the subtracks analyzed exhibit the qualities of a self-affine surface. Consequentially, we conclude that the Antarctic Ice Sheet surface is largely self-affine.

Figure 2 also shows little spatial variation of the Hurst exponent across the ice sheet. The average Hurst exponent across the continent is ≈0.41 with 95% of Hurst exponents estimated to be between 0.35 and 0.48. Floating ice has an average Hurst exponent of 0.47, and grounded ice has an average Hurst exponent of 0.41 (Supplementary Fig. 4). This difference between the mean of the two distributions is statistically significant as assessed by Welch's t-test, with a reported $P$ value of 0. With higher mean Hurst exponent, ice shelf surfaces have larger scale roughness features, since basal melt rates tend to be higher

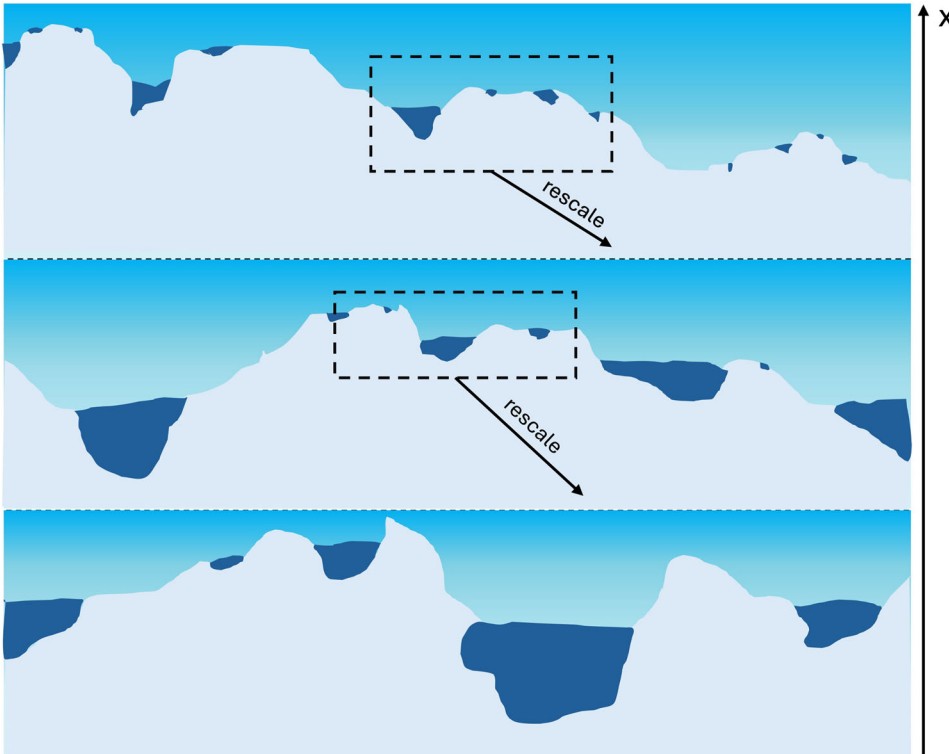

**Fig. 1 | The self-affinity of a glacier surface.** Illustration demonstrating the relationship between the vertical and horizontal scales of a glacial self-affine surface. The horizontal length scale decreases moving down in the Figure. Blue areas are supraglacial melt lakes filling local depressions. This figure is inspired by Figure 1 of [22].

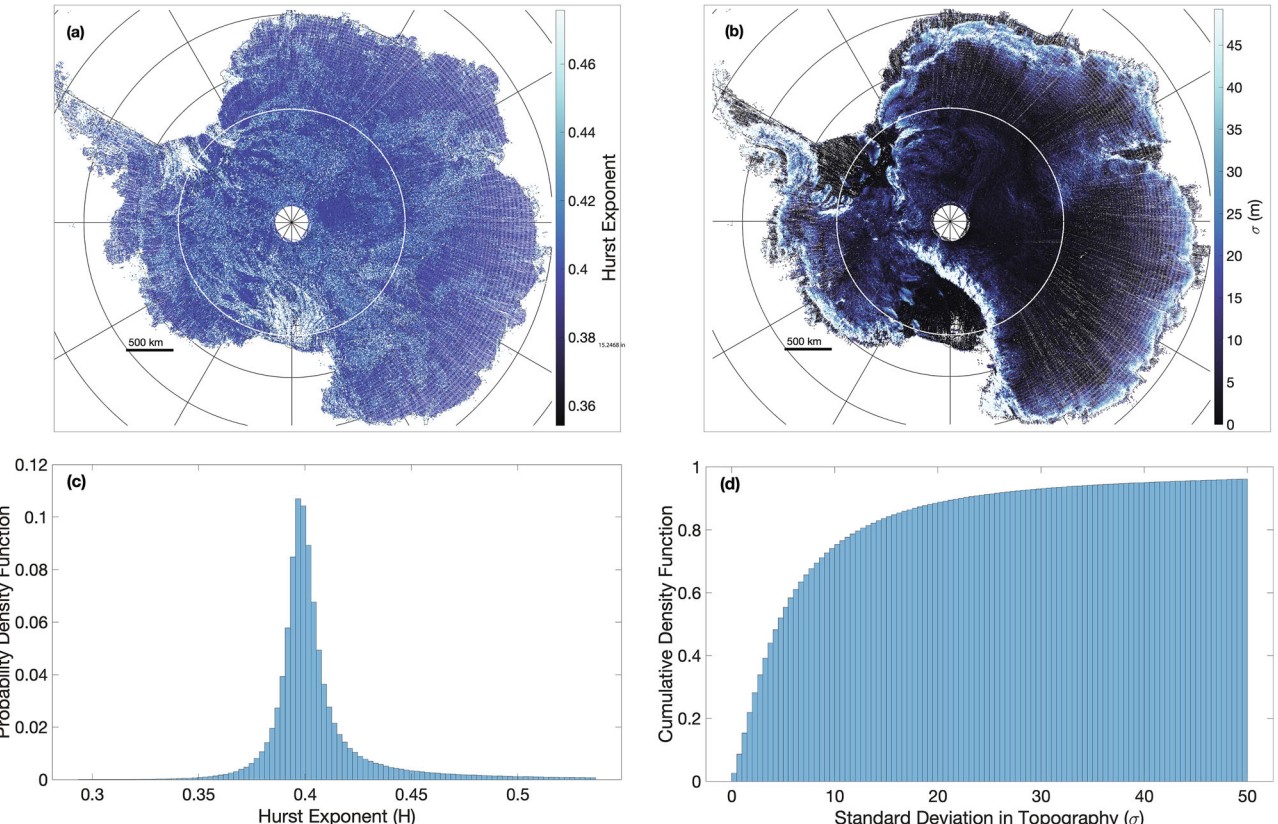

**Fig. 2 | Surface roughness quantification of the Antarctic Ice Sheet. a** The calculated Hurst exponents across land ice in Antarctica within a single deviation of the mean. **b** Standard deviation of topography across Antarctica. **c** The probability distribution function of the calculated Hurst exponents over Antarctic land ice. **d** Cumulative Density Function of the standard deviation of topography.

on steep small-scale basal roughness features[28,29] and the buoyant flexure of floating ice filters small scale roughness at the base from surface expression. The Hurst exponent is less than previously reported Hurst exponents for subglacial topography in Greenland as observed via ice penetrating radar in[25], which has a mean of 0.65. This indicates either a consistent difference between roughness in Greenland and Antarctica, or that processes related to snow redistribution and densification may influence surface topography, which creates smaller-scale topographical variability at the ice sheet surface than at the base[30].

The standard deviation, $\sigma$, quantifies the average amplitude of roughness along the ice sheet surface within each ICESat-2 sub-track and is plotted in Fig. 2b for the whole of the Antarctic ice sheet surface. $\sigma$ mainly varies over grounded ice due to the varying subglacial bed topography, over which ice flows. Greater roughness is notable over subglacial topographic features including the Trans-Antarctic mountains, Palmer Island, and Roosevelt Island. 74% of sub-tracks have relatively smooth ice sheet surfaces with $\sigma < 10$ m. Grounded ice has an average $\sigma$ of 13.0 meters, which is considerably rougher than floating ice with an average $\sigma$ of 4.7 meters (Supplementary Fig. 5).

We also investigated whether the estimated roughness parameters are biased by the track orientation or "capture" angle of the ICESat-2 satellite. We found that the Hurst values do not vary much based on the orientation angle of the satellite, and most of the measurements center around the value of  -0.4, which the average difference in Hurst exponent values at intersection points being 0.0218. For $\sigma$, the values between the two different sets of capture angles were also found to center around a single value. Details of this crossover analysis can be found in the supplementary material (Supplementary Fig. 1).

## Parameterizations for mean lake depth and area fraction on self-affine surfaces

In the prior section, we confirmed that roughness on nearly the entirety of the Antarctic ice sheet surface is self-affine. Since water gathers in surface depressions, the statistics of roughness on the surface can be used to determine the statistics of lake size on that surface, assuming that the surface topography does not change temporally[22,31]. While ice sheet surface melt may deepen depressions in which melt lakes form over many years, at first order, the pre-existing topography plays a strong role in determining melt lake size as validation with observations confirms in the validation section. Prior work in the statistical physics community has tackled the related problem of percolation on uncorrelated potential surfaces[20], producing analytical predictions for the size of clusters (melt lakes in our case) using mean-field mathematical approaches. However, such analytical approaches have been shown[21] to be intractable for strongly correlated surfaces ($H > 0$; i.e., self-affine) and surfaces that are not completely flooded (i.e., at their maximum capacity for water retention), such as those commonly found on Earth. For self-affine and partially flooded surfaces, numerical methods are necessary to calculate the statistics of lake size.

In this section, we use a cellular automaton (details described in "Materials and Methods"), a discrete algorithm that divides a larger surface into state-dependent cells, to simulate the progressive development of lakes on randomly generated self-affine surfaces. We run this cellular automaton for thousands of randomly generated self-affine surface, as a function of $\sigma$, $H$, and meltwater available for ponding (referred to as "supply" hereafter). From these numerical results, we calculate two melt lake size metrics, average area fraction ($\bar{F}$) and average supraglacial lake depth ($\bar{w}_l$). Once we have a set of model results encompassing the full parameter space, we seek to find a

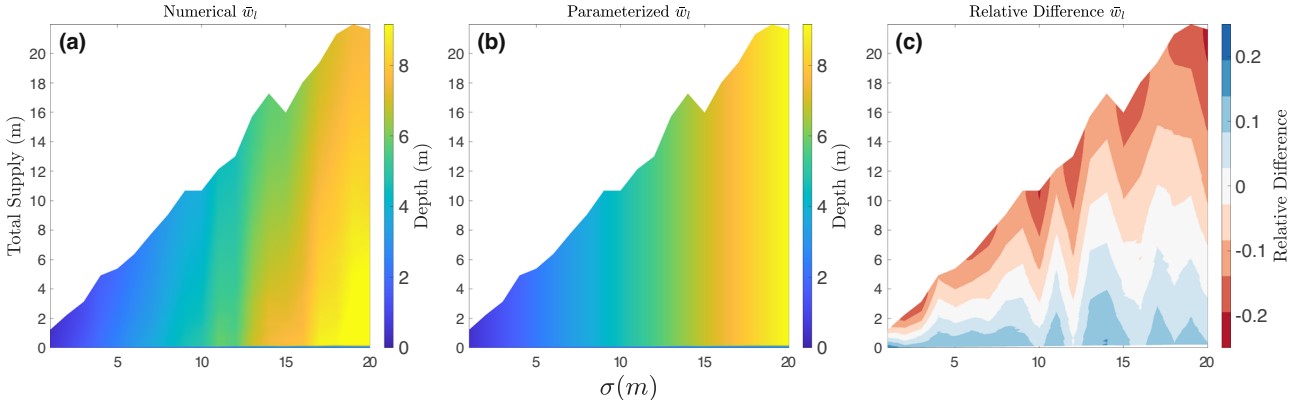

**Fig. 3 | Simulated and Parameterized mean supraglacial melt lake depth.**
**a** Parameter space of the mean lake depth-averaged over 500 randomly generated self-affine surfaces. **b** The parameter space of the mean lake depth of parameterizations fits with the input variables of the numerical simulations. **c** The relative difference between the numerical and predicted mean lake depth. The Hurst value is held constant at 0.4 for this parameter space.

simple mathematical relationship between each of the melt lake size metrics, and the three input parameters: surface roughness Hurst exponent ($H$), and amplitude ($\sigma$), and the average depth of meltwater supplied to the surface ($w_s$). However, first, we fit a relationship between the surface roughness parameters and the average depth of water over the entire surface when it is at its retention capacity, which we call $\bar{w}_d^*$. This quantity is frequently referred to as the "maximum depression storage" in terrestrial hydrology[32]. We calculate the maximum depression storage in a separate set of simulations using the full range of $\sigma$ and $H$ by provided an excess of water supply for the generated surface and extracting the mean depth across the entire surface. We then fit a simple mathematical expression that can determine the maximum average water depth of any self-affine surface at its maximum water retention capacity utilizing only surface roughness parameters:

$$\bar{w}_d^*(H, \sigma) = \sigma(0.2 - 0.12H^{0.6}). \tag{1}$$

Consistent with prior studies in hydrology and percolation physics[21,33], this equation predicts a linear increase (with a similar coefficient) in retained water depth with roughness amplitude and a sub-linear decrease in retained water depth with Hurst exponent.

We use this maximum depth to define $S$, a dimensionless "supply ratio"

$$S(H, \sigma) = w_s / \bar{w}_d^*(H, \sigma). \tag{2}$$

Defining this ratio normalizes differences between surfaces with strongly different retention capacities, and thus greatly improves the parameterization fit to numerical simulations over a wide range of roughness statistics.

We fit spatially averaged supraglacial melt lake size metrics to the topographical Hurst exponent ($H$), roughness amplitude ($\sigma$), and supply ratio ($S$). We know that average lake depth ($\bar{w}_l$) should be linearly dependent on $\sigma$ since the depth of lakes is set by the depth of depressions on the surface which are described by the roughness amplitude. The numerical results from the cellular automaton confirmed this linear dependency. By definition, the average horizontal extent of vertical topographic relief on a self-affine surface does not depend on the average roughness height[22]. Therefore, we expect that the area fraction of supraglacial melt lakes ($\bar{F}$) should have no dependence on $\sigma$. The numerical results from the cellular automaton also confirmed that $\bar{F}$ is not dependent on $\sigma$ despite the slight variation in Fig. 4(c) across the $\sigma$ space, which may be due to the sample size of surfaces leading to spurious residuals from the individual surfaces. With these results in mind, we were able to use standard least squares

regression tools to fit a set of two parameterizations that describe a relationship between each melt lake characteristic ($\bar{w}_l$ and $\bar{F}$) and the three parametric inputs ($\sigma$, $H$, $S$). The form of these equations is chosen to enforce certain basic physical constraints (described in "Materials and Methods"). We found the following parameterizations for supraglacial melt lake size metrics:

$$\bar{w}_l(H, \sigma, S) = 0.6\sigma \, \text{erf}\,(67S) \cdot (1 - 0.41H^{0.6}), \tag{3}$$

$$\bar{F}(H, S) = 0.13 \, \text{erf}\,(55S) \cdot (1 - 0.13H^{1.3} + 2\,\text{erf}\,(0.08S)). \tag{4}$$

These parameterizations all have an $R^2$ value greater than 0.82 compared to the generated numerical results from the cellular automaton. Figures 3 and 4 show the numerical predictions for melt lake metrics, the predictions from parameterizations, and the relative difference between the numerical results and parameterizations. The relative difference between the numerical and parameterized predictions is generally less than 25% for the lake depth parameterization and less than 10% for the area fraction parameterization. For both melt lake size metrics, the relative difference throughout the parameter space falls under 10% when the Hurst value is equivalent to 0.4, i.e., the mean Hurst exponent of the Antarctic ice sheet surface.

Two random self-affine surfaces with the same Hurst exponent and standard deviation can be quite different from each other and produce fairly different supraglacial melt lake distributions. Thus, when applying these parameterizations to a particular self-affine ice sheet surface area, they may depart from the melt lake size statistics due to random surface variation alone. In the supplementary material, we plot the root-mean-square deviation between the parameterizations and each realization from the numerical results, to quantify the expected natural variation among randomly generated self-affine surfaces (Supplementary Fig. 3). Since we have generated enough random self-affine surfaces in our Monte Carlo ensemble to reach statistical convergence in mean melt lake size metrics, the resulting parameterizations represent the typical size statistics expected for a given ice sheet area. Therefore, these parameterizations are useful for efficiently predicting spatially averaged melt lake metrics over large ice sheet areas. To predict the exact location and depth of melt lakes over a surface with known topography, a high-fidelity model of ice sheet hydrology should be used e.g.,[19], but at considerably greater computational expense which may be infeasible over larger spatial and temporal scales.

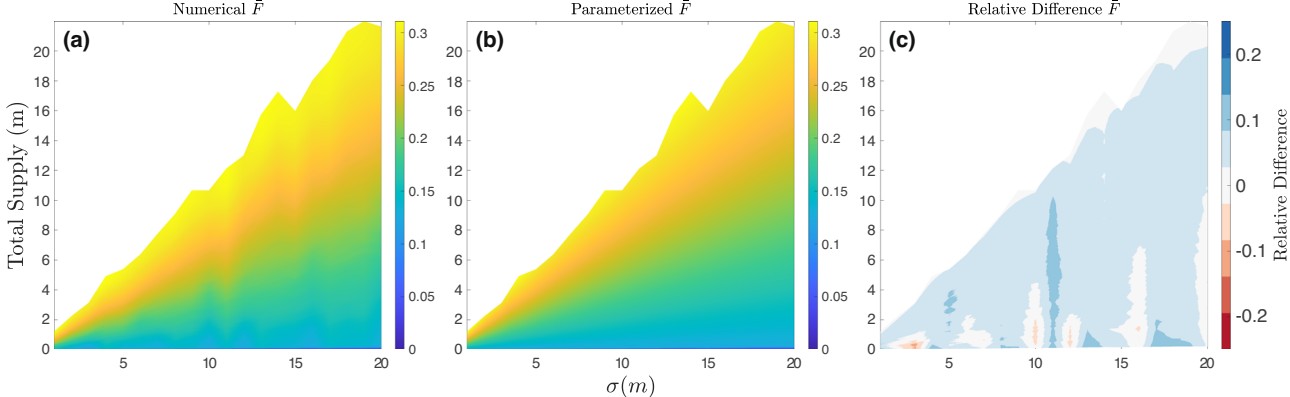

**Fig. 4 | Simulated and Parameterized mean supraglacial melt lake area fraction.**
**a** Parameter space of the mean area fraction-averaged over 500 randomly generated self-affine surfaces. **b** The parameter space of the mean area fraction of parameterizations fits with the input variables of the numerical simulations. **c** The relative difference between the numerical and predicted mean area fraction. The Hurst value is held constant at 0.4 for this parameter space.

## Validation of lake parameterizations with observations

After developing a set of physics-based parameterizations, we aim to validate the parameterizations against observations of melt lakes in Antarctica. To do so, we focus on two ice shelves with regular summer surface melt in the past 10 years: the Larsen C ice shelf in the Antarctic Peninsula and the Amery ice shelf in East Antarctica. To provide inputs to the parameterizations, we use measurements of $H$ and $\sigma$ for these ice shelves from the ICESat-2 analysis in the first Results subsection, and meltwater runoff supply as predicted by the RACMO regional climate model (described in "Materials and Methods"). We aim to validate our developed parameterizations by computing a range of mean melt lake depth and area fraction predictions for the Amery and Larsen C ice shelves, with inputs taken from the ICESat-2 derived topographical parameters ($H$ and $\sigma$) and the simulated meltwater runoff supply from the RACMO v2.3p2 model[34]. Due to the difference in time steps and spatial grid of the observational products and predictions, we include all ICESat-2 derived topographical parameters and RACMO seasonal accumulated runoff over the regions of each of the two ice shelves where melt lakes were observed in ref. 35. From RACMO meltwater runoff, we calculate the accumulated mean water runoff depth over the region where melt lakes are observed from the beginning of each melt season and then reset the accumulated supply to zero at the end of each melt season under the assumption that refreezing occurs. Additionally, we assume that the surface roughness has no changes and remains static over time. This results in a distribution of monthly predictions calculated from equations (3) and (4), with inputs taken over the area where supraglacial lakes are observed in Landsat, and including: surface roughness parameters estimated in the first Results subsection, and surface melt supply from RACMO. We compute the interquartile, 5th, and 95th percentiles of this distribution in each month over the years 2014–2019. This prediction assumes that the surface is impermeable and that no refreezing occurs during the melt season. Additionally, we assume that topographical parameters measured by the ICESat-2 mission in 2021–2022 apply to predictions over the 2014–2018 period when observations are available due to the relatively small temporal variations in these parameters during the ICESat-2 mission. Future validation exercises could use new melt lake observations derived directly from ICESat-2[36] to compare against predictions made from ICESat-2 topographical parameters. Here we use the[35] dataset to match the period covered by existing RACMO simulations. All further details of these validation exercises are detailed in the "Materials and Methods" section.

In Fig. 5, we plot observed (black dots) and predicted (blue shading) melt lake depth and area fraction at Amery and Larsen C ice shelves. Observed mean melt lake depth on both ice shelves (Fig. 5B) generally falls right within the 25%-75% range at the center of the distribution of predictions, though the melt season in RACMO seems to occur consistently later than the observed appearance of lakes (by 1–2 months). In contrast, observations of mean area fraction for both Larsen C and Amery ice shelves are consistently below the lowest quartile of observations. One possible reason for this over-prediction of the area fraction is that, in reality, meltwater may percolate into the snow and firn packs early in the melt season instead of flowing over the surface into lakes. Another possible explanation is that refreezing may still occur early in the season, thus preventing meltwater from accumulating at the ice shelf surface. As discussed above, any particular ice sheet surface may depart from the parameterization prediction due to random natural variability of self-affine surfaces, however, we consider melt lake metrics over a sufficiently large part of these ice shelves and determine that this is an unlikely explanation for these deviations. Additionally, the dependence of the area fraction parameterization on Hurst exponent is weak enough that it is unlikely that relatively small errors in roughness estimation from ICESat-2 can explain the validation mismatch. We further discuss these and other potential explanations for the mismatch between predicted and observed area fraction of supraglacial melt lakes in the Discussion section.

We also derive another set of predictions by inversely solving for the optimal water supply ($w_S$) to fit the observed mean area fraction using the mean Hurst exponent and mean standard deviation of topography for both Amery and Larsen C ice shelves. We scale the original runoff estimates from RACMO with a ratio of the aforementioned optimal water supply and mean runoff for each melt season under the assumption that either: some runoff percolates into the snow, some runoff refreezes, or RACMO overpredicts runoff. With this altered water supply, we perform the same calculations to develop an additional set of predictions (Fig. 6). Notably, the observations of area fraction primarily reside within the lower 25% of predicted area fractions. The opposite is true for the mean lake depth where predictions are near the lower end of observed lake depth at Larsen C, and are generally less than observed lake depths at Amery ice shelf.

## Maximum potential spatially-averaged Antarctic lake size under current roughness parameters

The maximum capacity of a rough surface to retain water is purely a function of roughness statistics[20,21]. As the water supply increases ($S \rightarrow \infty$), the size of supraglacial melt lakes, as captured in equations (10) and (11), eventually reaches a maximum:

$$\bar{w}_l^* = 0.6\sigma(1 - 0.41H^{0.6}) \tag{5}$$

$$\bar{F}^* = 0.13(3 - 0.13H^{1.3}) \tag{6}$$

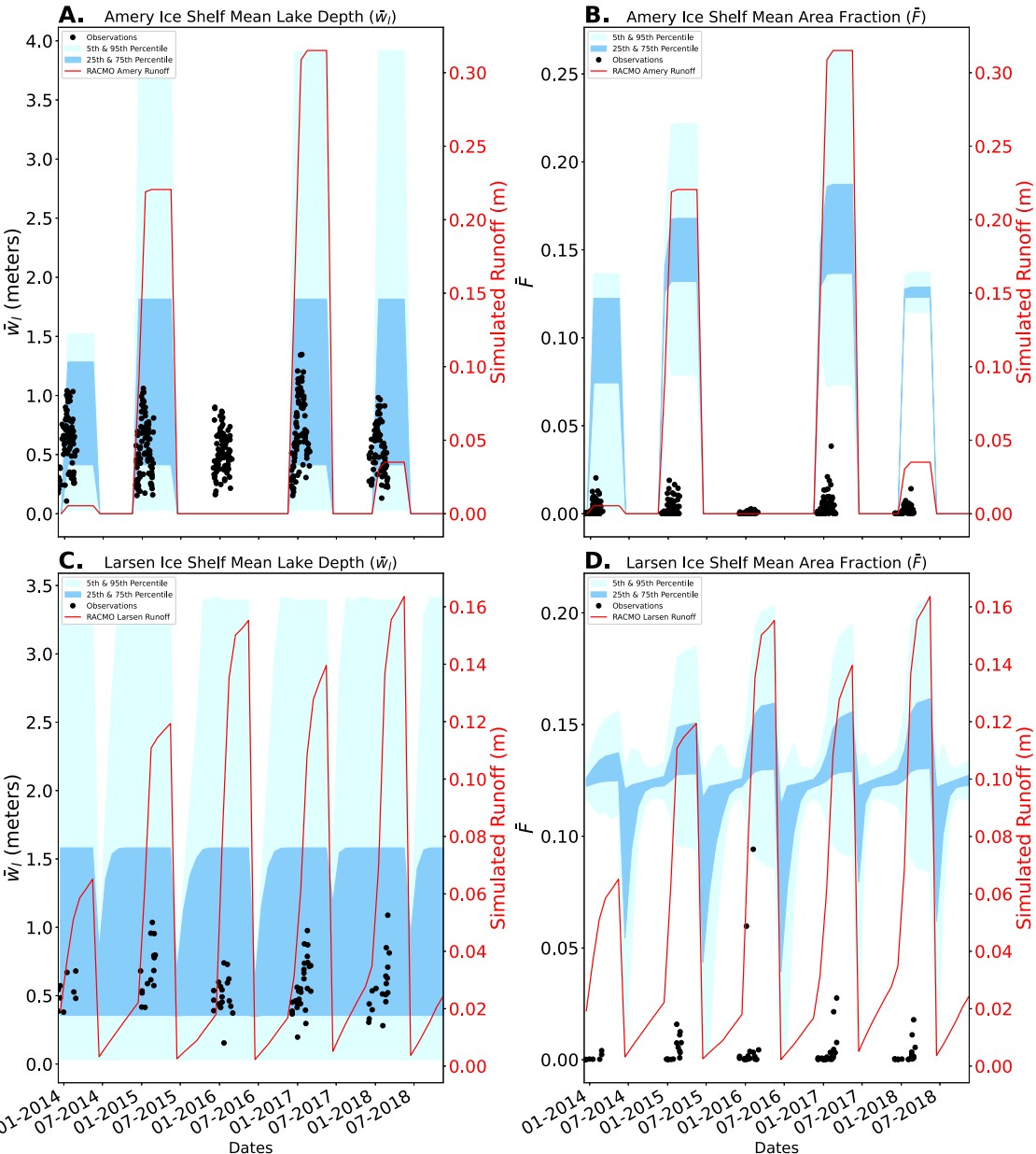

**Fig. 5 | Validation of parameterizations with Landsat-8 observational data from Larsen and Amery ice shelves. A, B** Predicted vs. Observed Mean Lake Depth and Mean Area Fraction for Amery Ice Shelf from 2013-2019. **C, D** Predicted vs. Observed Mean Lake Depth and Mean Area Fraction for Larsen C Ice Shelf from 2013–2019.

From equations (5) and (6), we see that the standard deviation of roughness ($\sigma$) directly influences the depth of water filling surface depressions. As $\sigma$ increases, the maximum spatially-averaged lake depth of the surface increases linearly. This is evident in the maximum-potential lake depth estimations mapped in Fig. 7 which largely resemble the map of roughness amplitude (Fig. 2b.), with a spatial average across the continent of 5.67 meters. As expected, ice shelves are predicted to have a lower maximum-potential lake depth, having a spatially averaged depth of 2.6 meters. Grounded ice has a maximum-potential lake depth of 5.9 meters averaged over space.

Lower Hurst exponents produce higher maximum area fraction of supraglacial melt lakes as shown at the Larsen C ice shelf (Fig. 8). This is due to a greater abundance of small-scale depressions in the topography. As the Hurst exponent increases, the maximum-potential area fraction decreases as there are fewer smaller depressions where melt gathers on the surface. As Antarctica has a narrow range of Hurst exponents, there is little variation in the maximum-potential area

fraction with a spatial average of 0.38 over the entire ice sheet area, which is the same when averaged over grounded ice or floating ice.

## Discussion

A more complex model that simulates more surface and hydrological processes may produce different results from the parameterized predictions here. Processes such as the percolation of meltwater through the firn and snowpack and refreezing of meltwater likely play an important role in determining the water available for lake formation at the ice sheet surface. The parameterizations developed in this study are first-order approximations of the relationship between surface roughness statistics and supraglacial melt lake characteristics. While these parameterizations may not have perfect accuracy in predicting what is found in observations, they can be used to improve our understanding and ability to easily include a representation of supraglacial melt lakes in large-scale climate and ice sheet models.

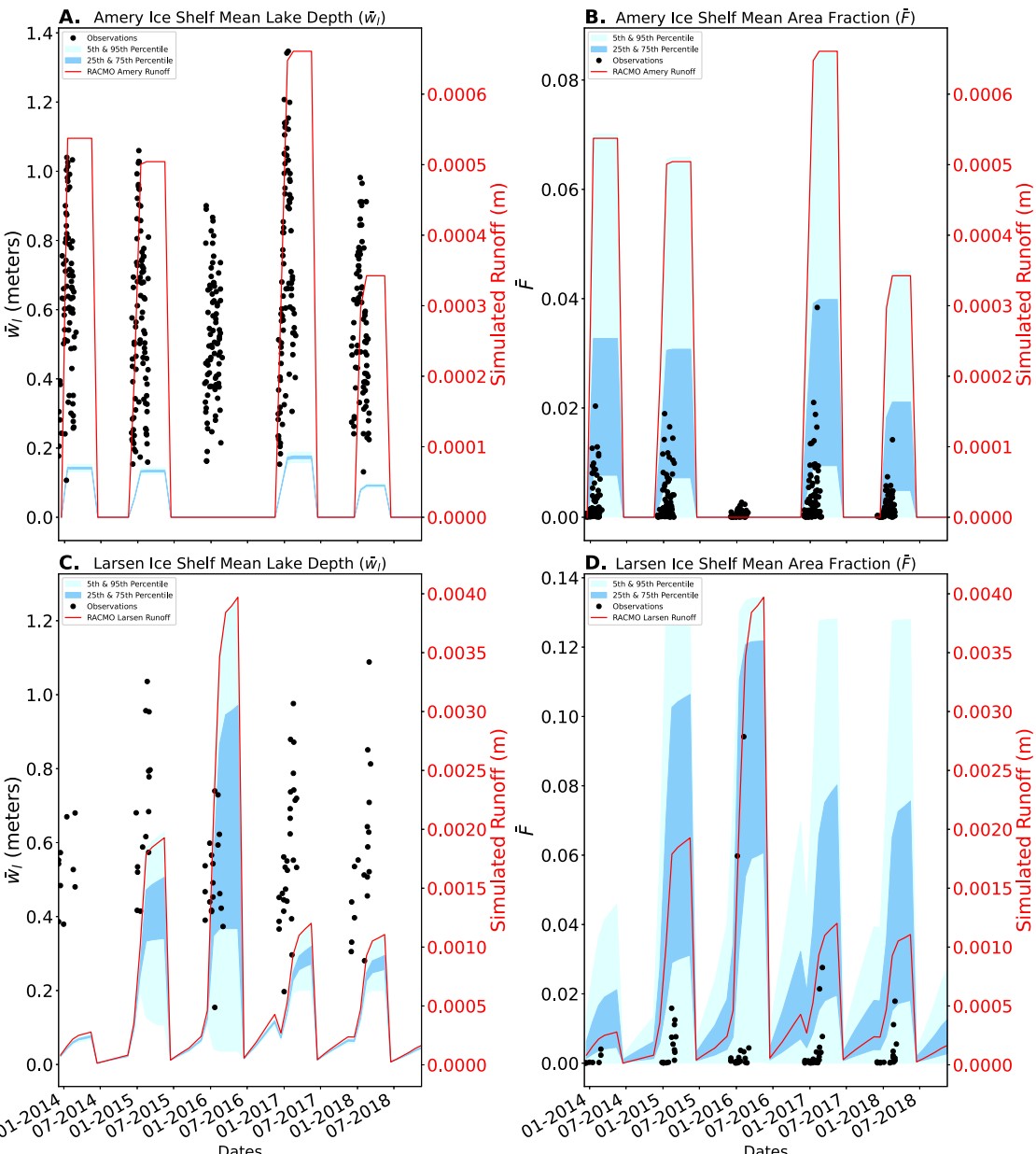

**Fig. 6 | Validation of parameterizations with an optimal melt water supply for the mean area fraction ($\bar{F}$).** Similar to Fig. 5, (**A**, **B**) display the predicted and observational mean melt lake depth and area fraction for Amery ice shelf, and (**C**, **D**) display the mean melt lake depth and area fraction for Larsen C ice shelf. These predictions are made with an altered water supply, derived from an inversion of the mean area fraction parameterization and RACMO runoff, with a bias to better fit the mean area fraction of both ice shelves.

As shown in the validation section, the parameterization of spatially averaged melt lake depth is consistent with observations at Amery and Larsen C ice shelves. However, the correspondence of area fraction to observations would likely be improved if these parameterizations were paired with a simple model of water storage in firn such as [37]. Limitations in the observational products that measure melt lake depth and area fraction used in the validation section could also potentially explain some of the mismatches. The current method for measuring these features is from satellite imagery and aerial imagery[35]. This method may not resolve smaller features such as slush and smaller or shallow lakes at the current functioning resolution and sensitivity of the sensors aboard imaging satellites. Visible satellite imagery is also known to underestimate lake depths due to saturation of red bands at low depths[38]. Another approach to validate these parameterizations could be to utilize a different observational platform, such as ICESat-2,

which can measure smaller surface features. Reference [39] compares lake depth measurements of the surface between ICESat-2 and satellite imagery (though a comprehensive ice-sheet-wide open-access product has not yet been released using ICESat-2 data) and concludes that using ICESat-2 improves the accuracy of the melt lake depth, with the satellite imagery underestimating the depth of the melt lakes. Future work could also repeat this validation exercise for melt observed on the Greenland ice sheet, which has more extensive, better-observed surface melt.

Our melt lake parameterization suggests that the albedo effect of melt lakes is not any greater on rougher ice surfaces than on smoother ones, since the area fraction of melt lakes does not depend on the surface roughness amplitude. Our parameterizations (and numerical water routing results) predict that once melt begins on a previously dry surface, melt lake depth quickly increases in the first 10's of cm of

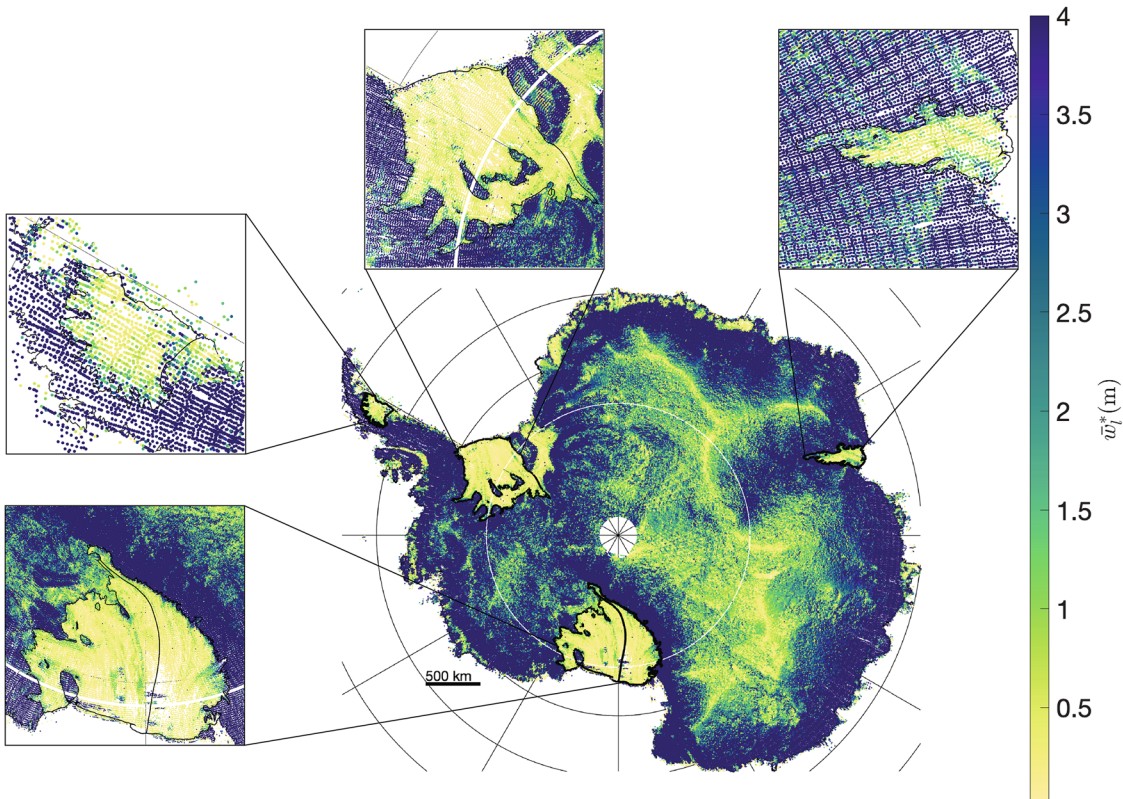

**Fig. 7 | Maximum potential supraglacial melt lake depth of Antarctica.** Map of mean maximum potential lake depth across the Antarctic Ice Sheet and ice shelves. Sub-figures show a closer look at the estimated mean maximum lake depth at Amery, Larsen C, Ronne, and Ross Ice Shelves. Most ice shelves have a significantly lower mean maximum lake depth than the ice sheet's interior.

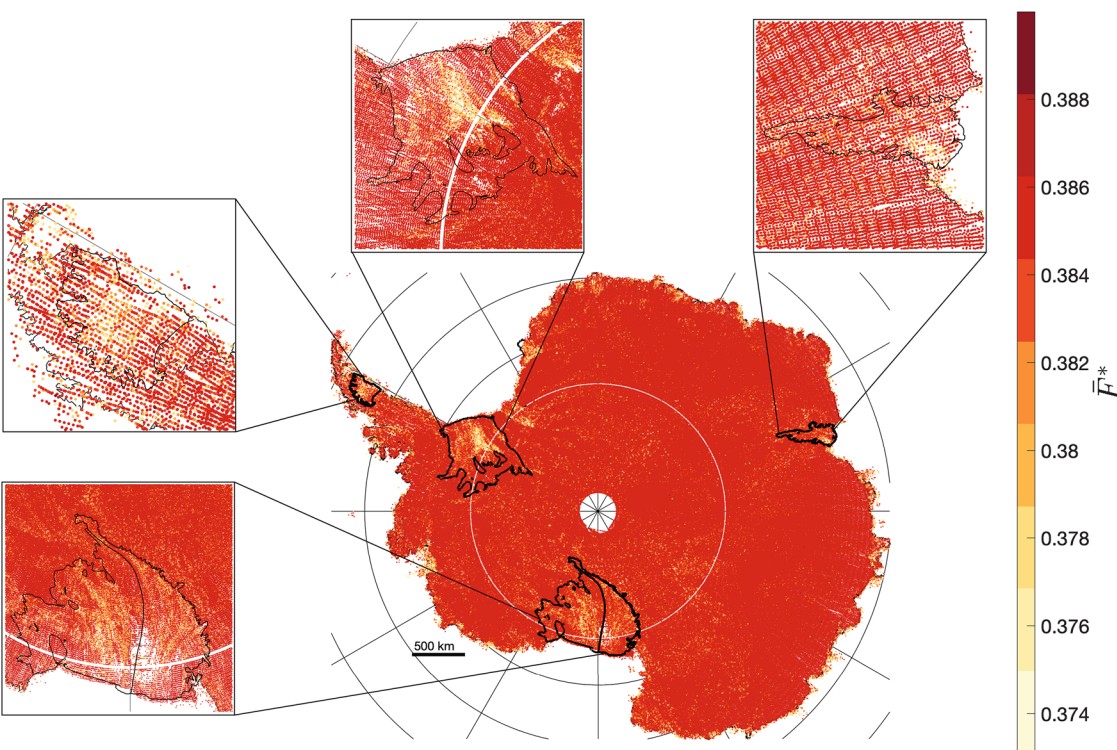

**Fig. 8 | Maximum potential supraglacial melt area fraction of Antarctica.** Map of the mean maximum area fraction a surface can have across the Antarctic Ice Sheet and ice shelves. Values for the mean maximum area fraction are uniform across the continent with some local variations observed on the ice shelves, which is largely responsible for its sole dependence on the Hurst exponent.

water supply, before reaching a maximal value at relatively low water supply levels. This may explain why even at relatively low levels of surface melt currently observed in Antarctica ( ~cm), melt lakes of several meters depth have been observed in specific regions[1,35,36]. Given the current ice sheet topography, the formation of such deep lakes may expand to new areas as surface melt spreads to these areas under future atmospheric warming, though the onset of melt lake formation may be delayed by percolation into available pore space in firn. However, our results suggest that the depth of existing lakes are unlikely to deepen further without increasing surface roughness. This may explain why, in the Amery and Larsen C ice shelf cases considered in the validation section, observed lake depths span a relatively narrow range, even under inter-annual variability in runoff meltwater supply. Still, this does not consider potential melt feedbacks that melt lake bottoms, causing them to further deepen through modification of surface topography.

Melt lake depth can confidently be predicted to be a linear function of surface roughness, indicating that rougher portions of the ice sheet surface are likelier to be locations of deep meltwater lakes, and therefore more susceptible to fracture propagation and glacier damage by hydrofracture. Parts of the ice sheets that are already deeply crevassed, such as the near-front regions of Antarctic ice shelves in Fig. 2a, are indicated by higher roughness in ICESat-2 measurements, and thus are more likely to retain ponded meltwater. However, it is important to also note that in some regions, the prevalence of surface fractures may also cause the surface to depart from self-affinity (i.e., Eastern Ross ice shelf in Fig. 2).

Given the relatively minor variation in Hurst exponents across the Antarctic ice sheet (Fig. 2), the variation in melt lake characteristics due to this factor is not expected to be significant. To a good approximation, the Hurst exponent in our derived parameterizations can be assumed to be 0.4 for grounded ice and 0.47 for floating ice. Adopting a single value of Hurst exponent will cause at most 5% error in the predicted melt lake size metrics.

In[25], the self-affinity of bed topography in northern Greenland is calculated, finding the average Hurst exponent to be ~-0.65, though with more spread across space than we estimate for the entirety of Antarctica. Besides the Hurst estimates of the subglacial and supraglacial topography, there are many studies estimating the Hurst exponent for terrestrial geomorphic surfaces, such as mountains, lakes, and coastlines, generally finding them to be between 0.4 and 0.5[22,23,40]. Our estimate of the Hurst exponent is thus similar to many other geomorphic surfaces, but perhaps not Greenland subglacial topography. This perhaps indicates that surface processes, such as those related to random deposition and coarsening of snow topography, may set the Hurst exponent of the ice sheet surface, as has been indicated by prior work on snow microtopography[30]. It is notable that the classic Kadar-Parisi-Zhang (KPZ) model for the growth of rough surfaces[41] produces surfaces with a Hurst exponent of 0.4. Thus, future work may explore whether this is a viable model for the production of ice sheet surface topography.

In this study, we develop simple mathematical expressions that predict spatially averaged supraglacial melt lake area fraction and depth using readily observable or modeled quantities as inputs. Most current large-scale models either drain all surface melt as runoff or retain all of it as a uniform water sheet on the surface (e.g.,[3,42]). The parameterizations developed in this study provide a simple way to include a first-order quantitative representation of the influence of supraglacial melt lakes on climate and ice sheets. Specifically, the area fraction parameterization could be implemented within albedo and surface energy balance schemes in global or regional climate/SMB models. The lake depth parameterization could be applied to existing crevasse propagation and fracturing schemes in large-scale ice sheet models[43].

In this study, we sought to fit low-order mathematical parameterizations that could readily be incorporated into a wide range of existing numerical models. However, using a similar training dataset, machine learning could also produce accurate parameterizations, using either outputs from a numerical hydrology model as in this study or directly from observed elevation and melt observations. However, simple mathematical parameterizations while potentially less accurate, have the advantage of being easy to understand and implement in large-scale models.

Whilst there has been an increase in observations of surface melting in Antarctica, there are many more studies and observations of supraglacial melting in Greenland. In a future study, the parameterizations should be tested on the Greenland Ice Sheet and could shed light on how the parameterizations perform with a higher melt supply scenario. A similar analysis performed in the first Results subsection would enable the comparison of roughness characteristics across two different ice sheets.

## Methods

### ICESat-2 land-ice elevation data set (ATL06)

To quantify the roughness properties of the ice sheet surface, we use measurements from the ICESat-2 satellite altimetry mission. The ATL06 product from ICESat-2 provides surface elevation at high spatial resolution covering nearly the entire ice sheet (less a small hole of missing coverage around the South Pole in the ice sheet interior). ICESat-2 repeats a near-polar orbit, repeating each track over Antarctica every 91 days, with an along-track spatial resolution of 20 meters between elevation measurements. The ATL06 data product measures the land-ice elevation and is a processed version of the raw photon data which measures the travel time between the satellite and the Earth's surface. Each ATL06 file has elevation estimates for the six beams, three weak and three strong beams. The weak beam estimates are best suited for bright surfaces such as snow and ice and so are used in this study. We select the first available weak beam from each data file in order not to double count data from both tracks. For every ATL06 track analyzed, we filter points in the track that have been registered to have an elevation greater than $10^{30}$ meters, which account for the unphysical elevation values.

Each ICESat-2 ATL06 elevation track is divided into sub-tracks ($h$) 10 km in length to remove the influence of long-range ice sheet shape on the elevation calculation. Since almost all supraglacial melt lakes are less than 10 km in horizontal extent, ice sheet roughness below 10 km is the most relevant for setting the average area and depth of supraglacial melt lakes. Two surface roughness parameters are computed for each sub-track: the standard deviation of elevation ($\sigma$), and the Hurst exponent ($H$). The standard deviation of elevation ($\sigma$) measures the square-root of the variance across the elevation of the subtrack.

### Computing the self-affinity of an ICESat-2 altimetry track

The self-affinity of a surface can be quantified through the Hurst exponent, $H$, which for topographic data measures the relative importance of topographic variations at a range of horizontal length scales 1. Put another way, a surface, $z(x, y)$, is self-affine if it can be rescaled by some factor $s$ according to the following equation:

$$s^{-H} z(sx, sy) = z(x, y). \tag{7}$$

The Hurst exponent has a value between zero and one. Hurst exponents near zero indicate that topography at short horizontal length scales has similar vertical relief to topography at long horizontal length scales. Hurst exponents near one indicate that topography at short horizontal length scales has much less vertical relief than topography at long horizontal length scales (i.e., the topography is dominated by smooth, large-scale features).

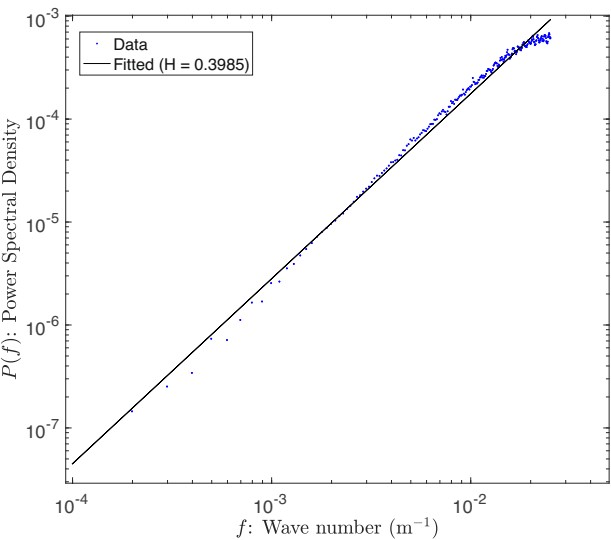

**Fig. 9 | Power Spectral Density analysis of an Antarctic ICESat-2 track.** The log-log plot of an ICE-Sat-2 elevation sub-track with an R² of 0.98. The $y$ axis inverts the logarithm of the power spectral density to make the power spectral coefficient ($\beta$) positive as shown in Equation (10). Blue markers indicated the calculated PSD values. The solid black line is indicative of the estimated linear regression. The slight deviation from the linear regression at higher wave numbers is due to the ATL06 product processing aggregate photon counts in 40 meters intervals[50], thus artificially smoothing roughness at the highest resolutions of the product.

We calculate the Hurst exponent using the "power spectral decay" (PSD) method of ref. 24. Other commonly used methods to calculate the Hurst exponent include Detrended Fluctuation Analysis[44] and Higuchi's method[45]. Here we find the PSD method to be computationally efficient and provides similar results to other methodologies used to compute the Hurst exponent[46,47]. In our approach, a Fast-Fourier transform ($\hat{h}(f)$) is applied to each sub-track of ICESat-2 ATL06 elevation data ($h(x)$), where the frequency is the spatial wavenumber along track ($f$)

$$\hat{h}(f) = \int_0^{10km} h(x)e^{-2\pi i x f}\,dx. \tag{8}$$

The power spectral density is defined as

$$P(f) = |\hat{h}(f)|. \tag{9}$$

We then fit a power law relationship between the power spectral density ($P(f)$) and the spatial wavenumber ($f$) by linear regression of $-\log(P)$ against $\log(f)$. Fig. 9 The slope of this linear regression is known as the power spectral decay coefficient,

$$\beta = \frac{-\log P(f)}{\log f}. \tag{10}$$

This $\beta$ coefficient can be written as a simple algebraic function of the Hurst exponent ($H$)

$$H = 2 - \frac{5 - \beta}{2}. \tag{11}$$

**Monte-Carlo Supraglacial Melt Drainage Simulations**
Our primary goal in this study is to find simple mathematical expressions relating mean melt lake area fraction and depth to surface roughness characteristics and meltwater supply. Since we have shown

in the prior section that the Antarctic ice sheet surface is self-affine, we focus on such surfaces in developing parameterizations. To achieve this goal, we follow a three-step method: (1) synthetically generate many random self-affine surfaces, (2) simulate water flow and ponding on these surfaces, (3) statistically fit simple mathematical relationships between mean melt lake area characteristics and surface roughness parameters and meltwater runoff supply.

We use a random self-affine surface generator which takes as inputs the Hurst exponent and standard deviation of the height profile. These inputs are used to calculate the power density spectrum and height probability distribution of surface[48]. As described in ref. 46, the surface is then generated from this height probability distribution through an inverse Fourier Transform method with randomized phases. Each generated surface has boundaries 10 km apart along a flat plain with topographic height with grid spacing of 100 meters.

After generating a surface, we simulate the flow of water into depressions on the surface using a "conditioned-walker" (CW) model developed for this study based on cellular automata[49]. The CW model takes as inputs: the surface elevation field and the total volume of meltwater supply. In the CW model described originally by ref. 49, a small discrete fraction (a "precipiton") of the total melt supply is added randomly at a location on the surface. The model moves the precipiton to one of the eight surrounding grid spaces with the lowest elevation and repeats this process until the precipiton reaches a grid space surrounded by grid spaces of higher elevation (i.e., a local minimum) or runs off the surface. This is under the assumption that the surface is non-porous with none of the precipitons penetrating the subsurface. The precipiton water thickness is now added to the surface elevation at this grid space until it equals the elevation of a surrounding grid space nearest in elevation. If water is still left over in the precipiton, then the movement process continues until no water is left. After a single precipiton has been deposited, a new iteration begins with an additional precipiton of water added to a random location on the surface, over a prescribed number of fill stages. This occurs until the prescribed total volume of meltwater has been distributed across the surface. At prescribed intermediate water supply levels, the CW model outputs a field of water depth.

The CW model has some features which make it ideal for considering water flow on relatively flat, rough, ice sheets. First, it allows excess water to flow off the edges of the surface, allowing for a surface's maximum meltwater capacity to be modeled realistically (in contrast to "depression-filling" water routing algorithms). Additionally, the CW model can simulate internally drained catchments that cause ponding on ice sheet surfaces, unlike many drainage algorithms used in terrestrial hydrology which assume that all water added to the surface is drained off the surface.

Utilizing the self-affine surface generation algorithm and the CW model, we simulate the distribution of water depth on many surfaces with a given Hurst exponent ($H$) and roughness amplitude ($\sigma$) and varied water supply ($w_s$). For each surface, we calculate the average melt lake depth ($\bar{w}_l$) over just the water-covered portions of the surface and the average area fraction of melt lakes ($\bar{F}$). Since the average melt lake depth ($\bar{w}_l$) is just the water depth only over the area fraction of the ice sheet surface covered by water ($\bar{F}$), we can, a priori, write a relationship between both characteristics and the mean water depth across the entire surface ($\bar{w}_d$)

$$\bar{w}_l \approx \frac{\bar{w}_d}{\bar{F}}. \tag{12}$$

Additional work is done to calculate the mean water depth across the entire surface in the Supplementary Material (Supplementary Fig. 2). Our workflow uses a Monte Carlo approach where all the melt lake metrics described above are averaged over 500 randomly generated surfaces to reduce the influence of single outlier surfaces. The result is

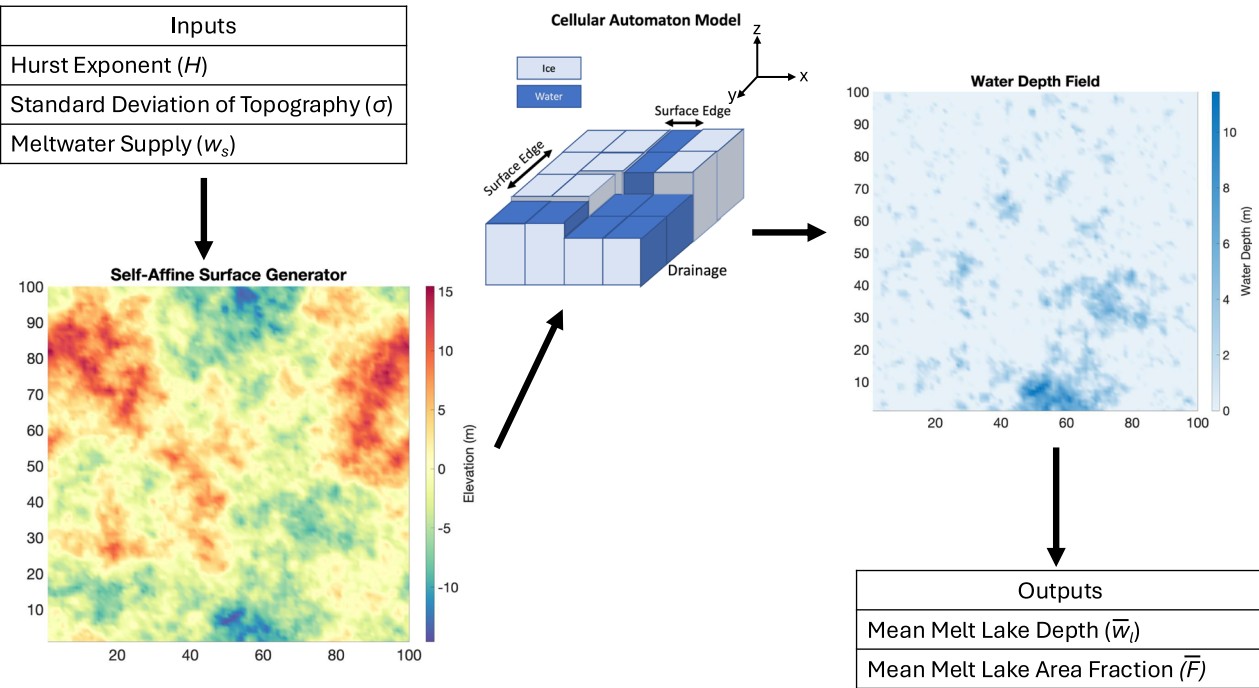

**Fig. 10 | Monte-Carlo workflow for supraglacial melt drainage simulations.** Schematic of the Monte-Carlo workflow to produce water depth fields which relate ice sheet roughness parameters to spatially averaged melt lake parameters.

a numerically generated dataset of average $\bar{w}_l$ and $\bar{F}$ values, each corresponding to inputs of $w_s$, $H$ and $\sigma$. This multivariate dataset is fit using standard nonlinear least squares fitting techniques to derive simple mathematical relationships between the melt lake characteristics. We set the functional form of equations for each melt lake characteristic to meet certain physical constraints (defined below), while using the simplest relationship between each independent parameter and the resulting melt lake parameters. In general, we prioritize forms of the parameterizations with relatively few parameters, and where no single term can be removed without substantially reducing the $R^2$ fit to the numerically generated dataset (i.e., simplicity). We also enforce two constraints:

$$\bar{w}_l(w_s = 0, H, \sigma) = \bar{F}(w_s = 0, H, \sigma) = 0 \quad (13)$$

and

$$\frac{d\bar{w}_l}{dws}\Big|_{w_s \to \infty} = \frac{d\bar{F}}{dws}\Big|_{w_s \to \infty} = 0. \quad (14)$$

The first constraint ensures that when there is no meltwater supply ($w_s = 0$), the melt lake has zero area fraction and depth. The second constraint ensures that all melt lake size statistics asymptotically approach a constant value since finite-scale bumpy surfaces have a finite capacity to store meltwater. To enforce the second constraint while obtaining smooth parameterizations, all terms involving the water supply are parameterized in terms of the error function, erf ($w_s$). We emphasize that there are likely other mathematical forms of these functions that would fit the numerical results equally well, but we search for mathematical functions that are relatively simple and easy to implement in large-scale models.

We generate surfaces with roughness varying over the full range of possible $H$ values ([0, 1] with $\Delta H = 0.1$) and a wide range of $\sigma$ values encompassing ~90% of estimates for the Antarctic Ice Sheet (1–20 meters with $\Delta\sigma = 1$m; see Fig. 2b). For each ($H, \sigma$) pair, we add meltwater

until the surface has reached its maximum water retention capacity (i.e., $\bar{F}$, $\bar{w}_l$ and $\bar{w}_d$ no longer change). For every possible ($H, \sigma, w_s$) value combination, 500 surfaces are generated and processed under the workflow described in the previous section, making a total of 110,000 simulations to make up the entire parameter space. A mean lake depth and area fraction are calculated for each ($H, \sigma, w_s$) parameter combination, averaging over each surface domain and all 500 generated surfaces. A schematic of the workflow can be seen in Fig. 10.

### Observations from Landsat
We validate our parameterizations with observations of melt lake area and depth from the Landsat-derived product of [35]. This product uses Landsat 8 Level 1 imagery that has been corrected to account for Top of the Atmosphere (TOA) reflectance and Sentinel 2 Level1C products to analyze the surface conditions of ice sheets and shelves[35]. Utilizes the Normalized Difference Snow Index (NDSI) to differentiate rocks from snow and the Normalized Difference Water Index to determine which pixels contain water. From this classification[35], computes the lake depth in each pixel by taking the difference between the reflection of optically deep water and the albedo of the lake bed and the top of atmosphere reflectance of the lake respectively and dividing the difference by the attenuation rate (Equation (15)).

$$z = \frac{\ln(A_d - R_\infty) - \ln(R_w - R_\infty)}{g}. \quad (15)$$

The area fraction is calculated as the ratio of the number of pixels classified as surface melt to the total number of land pixels, which are pixels classified either as surface melt or ice. This product has been developed for the entirety of Antarctica though here we focus on Larsen C and Amery ice shelves because: (1) this observational dataset indicates recurrent surface melt on these ice shelves in recent melt seasons, and (2) regional climate models also predict some surface melt on these ice shelves. Many regional climate models (e.g., RACMO

v2.3p2 which we use here) predict no surface melt runoff on many other Antarctic ice shelves where melt lakes are observed.

## Data availability

The processed data used in this study are available in the Zenodo repository under https://doi.org/10.5281/zenodo.15467941. NASA Icesat-2 ATL06 data used in this study is publicly available through the NSIDC. The melt runoff data is also publicly available[34], and the supraglacial melt lake observations is available within[35].

## Code availability

All code used to generate analysis Antarctic surface roughness, simulate melt distribution to develop the parameters, and to develop all the figures in this work can be found as a permanent Zenodo repository, which will be available upon publication. Scripts for numerical simulations are available on the following GitHub repository: https://github.com/dgrau13/meltlake-parameterizations.

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

## Acknowledgements

We would like to acknowledge S. Buzzard for her contributions to the discussion during the work of this project. This work and the lead author were funded by the NASA Modeling, Analysis, and Prediction Program, Award 80NSSC20K1131.

## Author contributions

This project was conceived by A.A.R. and D.G. D.G. wrote the first draft of the manuscript. D.G., A.A.R., and A.H. commented on and edited the manuscript. D.G. performed surface roughness analysis of the IceSat-2 Data, ran numerical simulations, and generated prediction figures within the validation. A.H. performed validation calculations for observed melt lake characteristics.

## Competing interests

The authors declare no competing interests.
