## [Transparent Peer Review file · Nature Communications]

Predicting Mean Depth and Area Fraction of Antarctic Supraglacial Melt Lakes with Physics-Based Parameterizations

Corresponding Author: Ms Danielle Grau

Version 0:

Reviewer comments:

Reviewer #1

(Remarks to the Author)

Summary

The paper focuses on the development of a parameterization for supraglacial lakes in Antarctica than can potentially be used by numerical atmosphere and glacier models to account for the effects of surface meltwater impoundment on the ice sheet energy balance and stability, respectively. The authors fit the equation for self-affine surfaces to ICESat-2 ATL06 elevations for Antarctica in order to empirically parameterize Antarctic surface roughness. They use the parameterized roughness coefficients to create synthetic ice sheet surfaces, which are then filled with water to varying levels to generate a sufficiently large dataset to develop their supraglacial lake parameterizations. The supraglacial lake parameterizations are assessed with respect to optical image-based estimates of supraglacial lake depth and fractional area coverage for the Larsen C and Amery ice shelves. When RACMO is used for the meltwater input to the parameterization, lake depths are reasonably reproduced but the fractional lake area is over-estimated. When the runoff inputs are tuned to better reproduce fractional lake area, lake depths are typically under-estimated.

Overall this is an interesting topic and the parameterization for supraglacial lakes is an important step forward in incorporating complex surface melt processes in numerical models without increasing computational effort/time. I think the explanation of the self-affinity and the Hurst equation are terrific. The writing is mostly clear but there are a few places where a more detailed description or an additional figure is necessary. Most of my comments are fairly general, but I have a few specific comments for typos and other small details.

Major Comments

1. The use of ICESat-2 elevations to characterize surface roughness is a really cool idea. You don't need to describe ATL06 in detail because it is used a lot and there are technical documents that the reader can review if they'd like. However, there are several data and methods descriptions in this section that require more detail due to the importance of the ICESat-2 data interpretation on your parameterization.

a. line89: "Depending on availability, all three of the weak beam elevation estimates are used." Does this statement mean that the ATL06 weak beam data were not always available because of clouds or something else?

b. line 90: "the entirety of the elevation profile of the track is analyzed to remove any elevation points that are a result of atmospheric back-scattering, which leads to unrealistic elevation values." You just mean that you used the ATL06 data product, which filters-out background signal photons, correct? You didn't do any additional filtering from what I can tell but this sentence doesn't clearly ascribe the filtering to the ATL06 data product.

c. lines 119-123: You state that you filter the tracks that don't support self-affine surface topography but describe the topography as self affine throughout the paper. Your filtering and interpretation of the ICESat-2 data would really benefit from a figure and a quantitative description of the filtering. How many (number and fraction) sub-tracks were filtered for Hurst exponents <0 or >1 ? How many were filtered based on the R^2 value of the linear regression to the log-log plot? If you added a figure in the main text that showed all the ICESat-2 tracks as lines with colors corresponding to each one of those filtering criteria, that would be really helpful because the reader would immediately get a sense of whether you have removed important data. Your other figures make it look like you have total coverage of Antarctica, so you could potentially

get by with simply reporting those metrics on filtered tracks. It's up to you but more detail should be provided.

2. For the Hurst equation parameters, is sigma always defined as such: "The standard deviation of elevation (sigma) measures the average root-mean-square deviation of elevation from the mean value within the sub-track."? Is this measure of variability typically prescribed using the standard deviation in elevations but you use something else here and simply keep the sigma notation to preserve convention? It's a bit confusing to call something the standard deviation but then not use the standard deviation.

3. For all your different plots (like figures 4-5), there is a strange oscillation in errors between sigma values. This does not seem to be discussed anywhere, unless I missed it, and it is worthwhile to include in the discussion at a minimum. Is this due to a relatively small sample size (500) for such a large parameter space?

4. The validation using remote sensing estimates of lake extent and depth makes it seem like the modeled lakes are too shallow and wide. The remotely-sensed area estimates are more accurate than the depth estimates, correct? That should be mentioned in the discussion. A disagreement with remote sensing data doesn't necessarily mean your results are wrong and that needs to be articulated more clearly. You mention that Fricker et al (2021) compared imagery- and ICESat-2-based lake depths. Were the ICESat-2 lakes deeper than suggested by imagery? You could really strengthen your results with a bit more discussion here.

Line-by-line Comments

- Lines 31 & 53: I would not make surface energy balance an acronym. You don't use it enough to warrant it.
- Lines 31-43: This paragraph starts to wander a bit. When you get into the latent and sensible energy fluxes, it is not really clear what you want the reader to get out of the paragraph. Is the point of the paragraph to explain that supraglacial melt influences air-ice sheet interactions? I think you could streamline this a bit.
- Lines 47 & 49: References are needed for these statements.
- Lines 184-185: The use of "surface area" is a bit tricky here. I think you mean the grid is 10km by 10km with 100m grid spacing, but this paper focuses on surface roughness and that roughness inherently increases the surface area relative to a perfectly flat surface. I'm not sure how to rephrase it, but think about it.
- Lines 234-235: These two lines are confusing. You describe each H-sigma pair filled to capacity but then say you create 500 surfaces for each H-sigma-w combination. Is the filled H-sigma combination just an end-member? Do you create 500 other surfaces or 500 total? Or do you create 500 surfaces for each H-sigma-w combination that is possible (which would be thousands or more simulations)?
- Lines 314-321: Be careful with the phrasing here. You initially state that you assume refreezing occurs in the winter then that no refreezing occurs in the summer. This isn't contradictory but it makes it hard to decipher if you evolve the surface topography or leave it static over time.
- Line 318: Change to "each monthly RACMO output time step" or something similar.
- Figure 6: Panel B is incorrectly labeled. The caption should also make it clear what circles vs shading mean.
- Section 5 and elsewhere: For each sigma, you look at the "maximum mean lake depth prediction". Is this the spatial average for a given H-sigma paired that is filled to capacity? Or are multiple surfaces averaged? If a single surface is used, I'd describe it initially as "spatially-averaged maximum lake depth" even though that is longer. The combination of multiple ambiguous descriptors makes it difficult to read. You also switch the order of "mean" and "maximum" repeatedly throughout. Please check.
- Lines 385-387: How is this possible if there is no water penetration or drainage? As supply increases, do you just get more overtopping of disconnected water bodies so that the area expands without much of a change in water depth?
- Supplement line 14: I think you mean "inert" or "intrinsic" or something along those lines, not "inane".

(Remarks on code availability)

There is very little documentation with the code. Although there is technically a README, it doesn't contain any information. I did not try to reproduce the code.

Reviewer #2

(Remarks to the Author)

Please see attached PDF

(Remarks on code availability)

Version 1:

Reviewer comments:

Reviewer #1

(Remarks to the Author)

Review of "Predicting Mean Depth and Area Fraction of Antarctic Supraglacial Melt Lakes with Physics-Based Parameterizations"

by Grau et al.

submitted for consideration in Nature Communications

Summary

The authors implemented relatively modest revisions in order to address comments made by two reviewers during the

previous round of review. While many of the revisions improved the manuscript, I found that a few of them were not beneficial. Below, I call attention to the detrimental revisions and offered some recommendations for how they can be improved. A handful of other revisions are also included below.

Major Comments

1. You do not explicitly define the terms in Eqn. 2. For example, presumably h is the ICESat-2 elevation but that needs to be stated. Additionally, the bounds of the integration are listed as infinite but your sub-tracks are 10 km in length and it seems as though your frequencies are restricted to $10^{-3.5}$ - $10^{-9.2}$ (based on Fig. 2) but that I not defined or explained. Please provide more detail in the formulation of this equation.
2. In my opinion, Figure 1 does not help convey self-affinity. I always imagine snowflakes when thinking about fractals (see https://personal.math.ubc.ca/~cass/courses/m308-03b/projects-03b/skinner/ex-dimension-koch_snowflake.htm). I like the multi-scale approach, with the zoom insets, but each zoom should show the same level of complexity but at different scales. I imagine something in which large chevron-shaped lakes are visible at the largest scale and you zoom in once or twice to show that there are also smaller chevron-shaped lakes that are not as visible at the larger scales. That would be much more effective.
3. For Figures 8 and 9, the color scheme does not appear correct based on the legend. Based on my interpretation of these figures, the darker blue color brackets the 25-50th percentiles and the lighter blue brackets the 5-95th percentiles. The legends suggest the opposite.
4. I appreciate the inclusion of two different predictions for the validation regions but I think that the discussion of why either the area or depth parameterization is in error is still lacking. One of the major discussion points that is missing is a discussion of the elevation data sub-track size and the frequency range used for solving the Hurst values. In Figure 2, you can see that the slope of the line is not the same across the full range of frequencies under consideration. If you focused on frequencies from $\sim 10^{-5}$ to $\sim 10^{-3}$ only, you would have a much shallower slope for that best-fit line. You mention the influence of the Hurst exponent on results in lines 382-388 but that discussion tries to argue that Antarctica has little variation in the exponent and does not mention how your interpretation of the data could potentially influence the agreement with observations.

Line-by-line Comments

- Lines 20-21: The new sentence added at the end of the abstract is somewhat awkward and does not accomplish the goal of articulating the impact of the work. Please revise further to circle back to the first two sentences that focus on albedo and ice shelf stability. Your penultimate sentence describes potential limits on pond size and depth. What are the potential implications related to albedo and hydrofracture?
- Lines 91-92: This sentence still reads as though you only select certain weak beams because you cannot expect the reader to know that sometimes data is not available from all three beams. Please rephrase to indicate you use all available weak beams.
- Lines 122-129: Why not include the same level of detail about filtering as you described in the response to the reviewers?
- Line 203: You do not allow percolation into the subsurface so I would not use the term “percolation” to describe movement of water across the surface.
- Line 231-232: I agree that you need to explicitly state that you do not permit percolation into the subsurface but this statement should come in the previous paragraph.
- Line 306 and onwards: Why is the “s” in Landsat capitalized here when it is typically not in literature?
- Lines 307-318: I like the additional detail that was added but this paragraph could be condensed. Sentences 2-4 essentially repeat the same points but with slightly different details.
- Lines 336-337: Please rephrase. The revisions to this sentence did not improve readability.
- Lines 416-417: I would start this paragraph with “Our melt lake parameterization suggests...” or something along those lines. Currently it is unclear if you are making a statement about albedo based on your work or you should be referencing literature.
- Line 422: Broken reference.
- Line 477: That Github repository link is broken.

(Remarks on code availability)

The link took me to a 404 error page.

Reviewer #2

(Remarks to the Author)

I appreciate the work that the authors put into the second version of this manuscript. The methodology is much clearer and I appreciate the addition of Figure 5. I agree with all of the changes made and only have a few minor additional comments/remarks:

Figure 1 – I’m not really sure what I am supposed to get out of the Figure. I think some labels and a more descriptive caption would be helpful. What are the blue spots? Also, I don’t think this Figure is referenced in the text (please check this for other figures as well)

In Figure 5 – does the height of the cells in the cellular automaton model indicate their relative elevation? It might be helpful to label the ‘z’ axis in this figure.

L272 – The sentence beginning with “As the depth of the melt lake...” does not make sense

L317 – Include space after equation

Figures, in general – Many of the figures have random horizontal and vertical lines around parts of the subfigures (eg. Figure 3, 11). This could be just in the rendering of the PDF but perhaps check to make the figures cleaner.

(Remarks on code availability)

Version 2:

Reviewer comments:

Reviewer #1

(Remarks to the Author)

The authors appear to have addressed all comments from the last round of revisions.

(Remarks on code availability)

Reviewer #2

(Remarks to the Author)

I appreciate the changes made to Figure 1 in the manuscript. Overall, I am happy with the modifications that the authors have made and am happy to accept the article for publication.

(Remarks on code availability)

We sincerely appreciate the constructive feedback provided by the reviewers. We have addressed all suggested edits. Below you can find reviewer comments in blue and our responses in black.

Reviewer 1

Summary

The paper focuses on the development of a parameterization for supraglacial lakes in Antarctica that can potentially be used by numerical atmosphere and glacier models to account for the effects of surface meltwater impoundment on the ice sheet energy balance and stability, respectively. The authors fit the equation for self-affine surfaces to ICESat-2 ATL06 elevations for Antarctica in order to empirically parameterize Antarctic surface roughness. They use the parameterized roughness coefficients to create synthetic ice sheet surfaces, which are then filled with water to varying levels to generate a sufficiently large dataset to develop their supraglacial lake parameterizations. The supraglacial lake parameterizations are assessed with respect to optical image-based estimates of supraglacial lake depth and fractional area coverage for the Larsen C and Amery ice shelves. When RACMO is used for the meltwater input to the parameterization, lake depths are reasonably reproduced but the fractional lake area is over-estimated. When the runoff inputs are tuned to better reproduce fractional lake area, lake depths are typically under-estimated. Overall this is an interesting topic and the parameterization for supraglacial lakes is an important step forward in incorporating complex surface melt processes in numerical models without increasing computational effort/time. I think the explanation of the self-affinity and the Hurst equation are terrific. The writing is mostly clear but there are a few places where a more detailed description or an additional figure is necessary. Most of my comments are fairly general, but I have a few specific comments for typos and other small details.

Major Comments

1. The use of ICESat-2 elevations to characterize surface roughness is a really cool idea. You don't need to describe ATL06 in detail because it is used a lot and there are technical documents that the reader can review if they'd like. However, there are several data and methods descriptions in this section that require more detail due to the importance of the ICESat-2 data interpretation on your parameterization.

a. line89: "Depending on availability, all three of the weak beam elevation estimates are used."

Does this statement mean that the ATL06 weak beam data were not always available because of clouds or something else?

As listed in the ATL06 Version 006 known issues document, there are granules that were available in previous releases are no longer available in later releases such as 006, which we use here. Line 89 was appended to the following: "Depending on availability on the 006 version of ATL06, at least one of the three of the weak beam elevation estimates are used."

b. line 90: "the entirety of the elevation profile of the track is analyzed to remove any elevation points that are a result of atmospheric back-scattering, which leads to unrealistic elevation values." You just mean that you used the ATL06 data product, which filters-out background signal photons, correct? You didn't do any additional filtering from what I can tell but this sentence doesn't clearly ascribe the filtering to the ATL06 data product.

We did some filtering of the ATL06 product to ensure that there are no unphysical values in the land height elevation track. For instance, there have been reported land elevation values up to 10^{30} within an

ICESat-2 data track file. This is non-sensical as this would be five times thicker than Earth's atmosphere. Line 90 was appended to address that we filtered ATL06 elevation tracks.

c. lines 119-123: You state that you filter the tracks that don't support self-affine surface topography but describe the topography as self-affine throughout the paper. Your filtering and interpretation of the ICESat-2 data would really benefit from a figure and a quantitative description of the filtering. How many (number and fraction) sub-tracks were filtered for Hurst exponents <0 or >1 ? How many were filtered based on the R^2 value of the linear regression to the log-log plot? If you added a figure in the main text that showed all the ICESat-2 tracks as lines with colors corresponding to each one of those filtering criteria, that would be really helpful because the reader would immediately get a sense of whether you have removed important data. Your other figures make it look like you have total coverage of Antarctica, so you could potentially get by with simply reporting those metrics on filtered tracks. It's up to you but more detail should be provided.

Two additional lines were added to the manuscript demonstrating that only 7.35% of the sub-tracks analyzed were tossed out of the total 1.8 million sub-track dataset. Of the 7.3% of sub-tracks that were tossed from the data set, 7.1% was due to poor R^2 fitting of the linear regression, and the remaining 0.25% was due to the linear regression producing a Hurst Exponent value that was outside of the acceptable range $[0,1]$.

2. For the Hurst equation parameters, is sigma always defined as such: "The standard deviation of elevation (sigma) measures the average root-mean-square deviation of elevation from the mean value within the sub-track."? Is this measure of variability typically prescribed using the standard deviation in elevations but you use something else here and simply keep the sigma notation to preserve convention? It's a bit confusing to call something the standard deviation but then not use the standard deviation.

Apologies for the confusion. We use the built-in standard deviation function in MATLAB to compute this parameter. Line 97 has been rewritten to clarify the exact mathematical calculation performed to calculate sigma.

3. For all your different plots (like figures 4-5), there is a strange oscillation in errors between sigma values. This does not seem to be discussed anywhere, unless I missed it, and it is worthwhile to include in the discussion at a minimum. Is this due to a relatively small sample size (500) for such a large parameter space?

This variation is related to weak nonlinear dependencies on water supply that are not possible to resolve in the fit without adding substantial complexity to the chosen mathematical expressions. We have opted to keep our expression simple, with the understanding that such high-order dependencies can be resolved by using the more complex numerical model. The fourth paragraph of the results section has been appended to provide a small discussion regarding this matter.

4. The validation using remote sensing estimates of lake extent and depth makes it seem like the modeled lakes are too shallow and wide. The remotely-sensed area estimates are more accurate than the depth estimates, correct? That should be mentioned in the discussion. A disagreement with remote sensing data doesn't necessarily mean your results are wrong and that needs to be articulated more clearly. You mention that Fricker et al (2021) compared imagery- and ICESat-2-

based lake depths. Were the ICESat-2 lakes deeper than suggested by imagery? You could really strengthen your results with a bit more discussion here.

Yes, the area estimates are more accurate than the depth estimates. In Fricker et al (2021) measures the depth of the melt lakes using ICESat-2 altimetry and compares them to the depths that were computed using satellite imagery. It was found that the satellite imagery derived depths underestimate the depth of the melt lakes. Both these points have been added to the discussion:

“Fricker et al. [2021] compares lake depth measurements of the surface between ICESat-2 and satellite imagery (though a comprehensive ice-sheet-wide open-access product has not yet been released using ICESat data) and concludes that using ICESat-2 improves the accuracy of the melt lake depth...”

Line-by-line comments

- Lines 31 & 53: I would not make surface energy balance an acronym. You don't use it enough to warrant it.

Eliminated all acronyms in the manuscript.

- Lines 31-43: This paragraph starts to wander a bit. When you get into the latent and sensible energy fluxes, it is not really clear what you want the reader to get out of the paragraph. Is the point of the paragraph to explain that supraglacial melt influences air-ice sheet interactions? I think you could streamline this a bit.

Reworded paragraph to focus on how supraglacial melt lakes effect the surface energy balance of glacier surfaces. Paragraph is now shorter and more streamlined.

- Lines 47 & 49: References are needed for these statements.

Lutz et~al. 2023 – measures the distribution changes of melt lake changes in Northeast Greenland

- Lines 184-185: The use of “surface area” is a bit tricky here. I think you mean the grid is 10km by 10km with 100m grid spacing, but this paper focuses on surface roughness and that roughness inherently increases the surface area relative to a perfectly flat surface. I'm not sure how to rephrase it, but think about it.

Rewrote the sentence to clarify that the horizontal bounds of the plain surface of 10 km apart with a grid spacing of 100 meters.

- Lines 234-235: These two lines are confusing. You describe each H-sigma pair filled to capacity but then say you create 500 surfaces for each H-sigma-w combination. Is the filled H-sigma combination just an end-member? Do you create 500 other surfaces or 500 total? Or do you create 500 surfaces for each H-sigma-w combination that is possible (which would be thousands or more simulations)?

We have revised these lines to specify the total number of simulations that end up making the parameter space. Each H-sigma pair can be used to randomly generate surfaces. Thus, we sweep over X values of H, Y values of sigma and, and for each pair of parameter values we randomly generate 500 surfaces. Thus, we run $X*Y*500=ZZZ$ simulations using the cellular automaton model. For each simulation, we consider output at N different levels of water supply.

- Lines 314-321: Be careful with the phrasing here. You initially state that you assume refreezing occurs in the winter then that no refreezing occurs in the summer. This isn't contradictory but it makes it hard to decipher if you evolve the surface topography or leave it static over time.

We have clarified this sentence by changing to "...reset the accumulated supply to zero at the end of each melt season under the assumption that refreezing occurs."

- Line 318: Change to "each monthly RACMO output time step" or something similar.

Fixed.

- Figure 6: Panel B is incorrectly labeled. The caption should also make it clear what circles vs shading mean.

Panel B has been corrected.

- Section 5 and elsewhere: For each sigma, you look at the "maximum mean lake depth prediction". Is this the spatial average for a given H-sigma paired that is filled to capacity? Or are multiple surfaces averaged? If a single surface is used, I'd describe it initially as "spatially-averaged maximum lake depth" even though that is longer. The combination of multiple ambiguous descriptors makes it difficult to read. You also switch the order of "mean" and "maximum" repeatedly throughout. Please check.

We have made changes in this section to refer to "Maximum potential spatially-averaged Antarctic lake size". This should clarify that we mean the first of your suggestions, the "spatial average for a given H-sigma paired that is filled to capacity".

- Lines 385-387: How is this possible if there is no water penetration or drainage? As supply increases, do you just get more overtopping of disconnected water bodies so that the area expands without much of a change in water depth?

Yes, this is correct. Every surface has a maximum water retention (Schrenk et al., 2014) regardless of percolation and maximum saturation of the firn. As water is added to the already saturated surface, excess water will flow off the surface into neighboring cells and surfaces until they are fully saturated as well.

- Supplement line 14: I think you mean "inert" or "intrinsic" or something along those lines, not "inane".

Fixed.

Reviewer #1 (Remarks on code availability):

There is very little documentation with the code. Although there is technically a README, it doesn't contain any information. I did not try to reproduce the code.

GitHub was updated for all scripts including the README, which now has a description of every script in the depository. Updated GitHub can be found in the manuscript as well as here.

<https://github.com/dgrau13/meltlake-parameterizations>

Reviewer #2

Review of “Predicting Mean Depth and Area Fraction of Antarctic Supraglacial Melt Lakes with Physics-Based Parameterizations”

Grau et al. 2024

Summary

In this manuscript, the authors provide parameterizations for supraglacial melt lake characteristics that could be used in future ice sheet modeling work. Overall, this is an interesting manuscript. It is well-written and the figures are generally clear. I have some major concerns relating to the methodology. Once these concerns are addressed, I believe this work will be a good contribution as parameterizations for supraglacial meltwater will be increasingly important in ice sheet modeling.

One general comment: The structure of this manuscript is a bit odd. Sections 2-4 each have individual methods and results subsections. I don't necessarily mind this structure and I found it relatively intuitive and easy to follow, but I'm not sure that it aligns with the journal requirements. If the current structure is maintained, I would recommend using more verbose titles for sections 2.2, 2.3, 3.1, 3.2 and 4.3 so there are not multiple subsections titled simply “methods” or “results”. For example, Section 2.2 could be renamed “AIS self-affinity methodology” or something of the sort.

Major Comments

Generally, I find some methodological details lacking. I also believe that an overarching diagram or schematic of all the methods would be helpful.

Additional figures depicting the self-affinity of an icy surface and the methodological workflow described in Section 3 have been added.

Self-affinity (Section 2)

- what quality proves that a surface is self-affine? It is unclear to me from this work, what actually demonstrates that the AIS is self-affine. Is it the Hurst exponent that proves self-affinity?

A surface is self-affine if the scale of vertical relief with respect to horizontal scales can be described by a power law distribution. Thus, the fact that we can statistically fit more than 90% of the surface area of Antarctica to such a power law distribution indicates that the AIS surface is largely self-affine. This fact is now stated plainly with reference to these results in section 2.2.

- The paragraph from L119-123 is confusing. In this section, I thought you were trying to demonstrate that the AIS is self-affine. But in these lines, the manuscript mentions that tracks are discarded if they are not described as self-affine or that “the only sub-tracks considered are those in which the data strongly indicate roughness is self-affine”. How

can you demonstrate self-affinity if you self-select only tracks that are self-affine?
Additionally, I find some details to be missing. For example: What percentage of the sub-tracks are selected? How many have $R^2 < 0.7$?

We have modified this section to be much clearer:

“Of the 1.8 million sub-tracks that were analyzed, 7.3% of the sub-tracks were discarded during the analysis process. As a result, over 92% of all the subtracks analyzed exhibit the qualities of a self-affine surface. Consequentially, we affirm that the Antarctic Ice Sheet surface is largely self-affine.”

Mean lake depth and area fraction (Section 3)

- L206 – “with varying water supply depth, Hurst exponent, and roughness amplitude”.

What do these values vary between? I guess this is in L232-233 but this information should come in the methods section. Also, more details are necessary. Are they randomly selected from the distributions found in Section 2 (i.e. Fig 2,3)? Can you elaborate on the “water supply depth”? Is this the same as the “meltwater availability”?

Where does this information come from?

We have modified this for clarity:

“Utilizing the self-affine surface generation algorithm and the CW model, we simulate the distribution of water depth on many surfaces with a set range of Hurst exponent (H) and roughness amplitude (σ) values and varied water supply (w_s).” Line 206 has been rewritten to specify that the water supply varies from a surface-to-surface basis. It also addresses that the range of the Hurst exponent and roughness amplitude is defined from a set range. This paragraph was also moved from the results section to the methods section expanding on the set-up of the model including the increment steps of the Hurst exponent and roughness amplitude of the parameter space. This water supply depth is merely the volume of the meltwater supply divided by the planar area of the generated surface. This water supply depth is equivalent to the meltwater supply or “runoff” as calculated in SMB models. Further clarification has been added.

- L232 – is it realistic to do this over the range of possible H values $[0,1]$ since it seems like >99% of the calculated H values fall between 0.3 and 0.6 (Fig. 2).

There are still some outlier values on the Antarctic Ice Sheet that fall outside of this range. We wanted to build a parametrization that could be applied to a variation of surface roughness conditions not one exclusive to the Antarctic Ice Sheet.

Validation of lake parameterizations (Section 4)

- Can you provide the equation used to calculate depth? Which bands are used? What are the limitations and uncertainties associated with these depth calculations?

Throughout this section it seems that you treat the ‘observed depths’ as the truth.

However, there are large uncertainties when using optical imagery to determine lake depth. A greater discussion of this is necessary.

Red and Blue bands are used to compute the Normalized Difference Water Index, which is able to detect whether a pixel contains water. We have added some general details about the calculation made in Moussavi et al. (2020), but the complete details for these calculations are in that original paper and therefore we opted not to repeat them in their entirety here. A further discussion of the uncertainties inherent in these calculations has been added.

- It appears from Figure 6 that for the lake depth and area fraction predictions, it is assumed that meltwater refreezes before July? In reality, melt refreezes much before then. Can you better temporally constrain the predictions? Maybe using RACMO air temperature?

This validation purely depends on the runoff estimates from RACMO. It would be challenging to attempt to constrain the predictions using the air temperature estimates, especially when attempting to scale the amount of depth that would be refrozen due to air temperature. An additional model or parameterization would be needed to implement this, unless we assume once the air temperature reaches a stable value below freezing that all the melt lakes freeze. Therefore, we have opted here to take RACMO outputs at face value as would be available in large-scale models with SMB simulated.

- Is there a way to include firn air content in this parameterization, thereby accounting for percolation?

There is a way to account for percolation in this parameterization by reducing the input melt water supply by assuming that “x” meters of melt water supply percolate into the firn leaving “x” meters of melt water supply to pool on the surface. This percolation could be considered to be dependent on firn air content. However, this is beyond the scope of our study, and indeed there are prior studies considering such percolation.

- L344 – what is the scaling factor used?

Modified to:

“We scale the original runoff estimates from RACMO **with a ratio of the** aforementioned **optimal water supply and mean runoff** for each melt season under the assumption that either: some runoff percolates into the snow, some runoff refreezes, or RACMO overpredicts runoff.” The optimal water supply is derived by solving inversely for w_s from the observational area fraction.

Discussion (Section 6)

- L369 –“The parameterizations developed in this study are mostly consistent with observations. ” I’m not really convinced by this. I am especially concerned with Figure 6b/d. Considering there are huge uncertainties in both the observations (lake depth from optical imagery) and the parameterizations (not accounting for firn percolation), I’m not convinced by the comparison to observations.

We have modified this to account for some of the shortcomings identified by the reviewer:

“Of the parameterizations developed in this study, the mean melt lake parameterization is mostly consistent with observations.” Ultimately, the two points brought up by the reviewer above are acknowledged and discussed in the text. We feel that the manuscript is stronger with a validation section than without, and absent more concrete suggestions about how we can improve the validation, we feel that it adds to the arguments made in this study.

- L390 – which areas specifically are most at risk of future lake expansion? Could there be some more advanced future insight included in this manuscript, besides just that other areas may be at risk of melt pond development?

Answering the question posed here is ultimately best left up to a more detailed regional climate model which can predict where runoff may occur. We do not purport to answer this question in our study, which has already been tackled by many other studies. Instead, we answer a question that has received much less

attention in the literature, namely: “If melt were to occur, what would be the size of melt ponds?” This question gets at different issues related to albedo and potential hydrofracture that previous studies focused on melt production are not able to answer.

- L392-393-“This may explain why, ... observed lake depths span a relatively narrow range” . Or perhaps, the algorithm used to calculate lake depth has saturated. Previous work (e.g. Lutz et al 2024) has shown that lake depths computed using the red band saturate at low depth, therefore a full range of lake depths cannot be calculated using this method. This limitation should really be discussed in comparison to observations.

We have added a discussion and reference to Lutz et al. (2024). Thank you for bringing this new study to our attention.

Abstract – I think you need a concluding sentence in the abstract that summarizes the motivation and importance of this work. This should tie back to the first 2 sentences of the abstract. As written now, the significance of this work is not immediately clear from the Abstract.

An additional sentence was added to the end of the abstract that restates the overall applications and limits of the parameterizations.

Minor Comments

L21 – I don’t think that the Kingslake et al. (2017) paper showed an increase in surface melting and supraglacial lake formation.

Removed increase so sentence now reads: “... observations have shown widespread surface melting and supraglacial lake formation on Antarctic ice shelves...”

L31-43 – This paragraph goes into a lot of detail about the impact of supraglacial meltwater on surface energy balance and energy fluxes. This seems like too much detail considering the manuscript is not about ice sheet surface energy balance. I would appreciate more background introduction on topography theory instead of energy fluxes.

The introduction is amended to have a small section discussing the attributes and properties of self-affinity and less detail on surface energy balance processes.

L48 –“Supraglacial melt lakes are omitted from most current large-scale models...” Are there any models that incorporate them?

This is correct, we have deleted the word “most”.

L53 – The last sentence of this paragraph should be reworded to be more specific and stronger regarding the contribution of this work.

We expanded the last sentence in this paragraph to be more direct with the expectations and applications to come from this work.

L55-59 – It would be helpful to have more background information here

The introduction is amended to have a small section discussing the attributes and properties of self-affinity and surface evolution.

L59-69 – Relate the work done back to the original knowledge gaps.

An overarching sentence was added describing the applications of this work and the intent.

L111 –“... provides similar results to other methods. ” Did you compare to other methods, or was this found in a previous study?

This was found in previous studies comparing different methods. Two citations were added at the end of the sentence for readers interested in learning more about the different methods and theory behind calculating the Hurst exponent.

L130 –“There is a clear difference in Hurst exponent between grounded and floating ice” . Is this difference statistically significant?

It is statistically significant. The p-value and t-test between both distributions are respectively zero and 1. We have added a statement to this effect.

L149 –“It was found that the Hurst values do not vary much...” How much? Can this be quantified in the main text?

We have added a statement indicating that the mean difference of the value of the Hurst exponents at the intersection point of the ICESat-2 track is 0.0218.

Figure 2,3 – Why do you show a PDF for the Hurst exponent and a CDF for the standard deviation?

To show that most of the values for the standard deviation of topography falls within 25 meters. There is a figure in the supplemental that demonstrates the PDF for the Hurst of the grounded ice and floating ice respectively, which exhibits a distribution like that of a Poisson distribution.

L167 – can you provide a brief description/explanation of “cellular automata”?

We have appended sentence to include a brief definition of cellular automata.

L174 –“meltwater runoff depth” This is the same as “meltwater availability” , right?

We have changed the phrase “meltwater runoff depth” to “meltwater supply”.

Section 3.1 – Somewhere in this section it is important to mention that these methods assume a non-porous surface and that there is no infiltration and all precipitons pond on the surface.

We added the following sentence:

“We also assumed that the surface is non-porous with none of the precipiton penetrating the subsurface.”

L216 – Can you elaborate what is meant by “custom equation forms”?

We have modified the sentence to read:

“We set the functional form of equations for each melt lake characteristic to meet certain physical constraints (defined below), while using the simplest relationship between each independent parameter and the resulting melt lake parameters.”

L234 – when writing the “ $H - \sigma$ pair” (and later for $H - \sigma - w!$), is there a better way to write this that does not look like subtraction? Maybe (H, σ)

Notation changed to (H, σ) and (H, σ, w_s) .

L245 –“... we create a simple mathematical expression...” How is this created? Can more detail be provided?

This part modified to read:

“ We calculate the maximum depression storage in a separate set of simulations using the full Hurst and standard deviation of topography range by over-saturating the generated surface and extracting the mean depth across the entire surface. We then fit a simple mathematical expression that can determine the maximum average water depth of any self-affine surface utilizing only surface roughness parameters.”

Figure 4/5 – Please ensure that all color bars are labeled.

We have added labels to all color bars.

Also, in 4a, for large σ , why is the average lake depth higher for lower total water supply?

The deepest depressions also tend to be the most laterally extensive (the result of self-affinity of the surface) and so these fill in first due to capturing more randomly generated surface melt, producing the deepest lakes. As further water supply is added small depressions are filled in, thus lowering the average lake depth. This is a relatively minor second-order effect and so we do not prioritize this particular feature in our simple parameterizations which are mainly meant to capture the initial rapid increase in lake depth at low water supply and the dependence of lake depth on sigma.

In Figure 5, the relative difference is almost always negative. This may indicate that F is not parameterized properly?

The relative difference is almost always positive in Figure 5. We have prioritized meeting the physical constraints of the parameterizations which produces some minor systematic error in this case.

L309 – Specify the “predictions” you aim to provide (e.g. mean lake depth, lake area fraction).

Modified to:

“We aim to validate our newly developed parameterizations by computing a range of mean melt lake depth and area fraction predictions for the Amery and Larsen C ice shelves...”

L314 – What is meant by “the regions within the two ice shelves where melt lakes are

Observed”?

Modified to:

“Due to the difference in time steps and spatial grid of the observational products and predictions, we instead include all ICESat-2 derived topographical parameters and RACMO seasonal accumulated runoff over the regions within the two ice shelves where melt lakes were observed in Moussavi et al., 2020.”

Figure 6 – what do the different shades of blue represent? Why is it often cut-off at the top of the plot? The y-axis should be extended to show the full range of predicted values. Also, would it be possible to include RACMO meltwater production on these plots?

Figure 6 has been redone to include a legend describing each component of the plot, and an additional axis added to incorporate the runoff (melt supply) for the validation.

L361 – When thinking about maximum lake depth, it does not make sense to me that “smaller more frequent bumps” would “correlate with a higher mean maximum water depth” . If the bumps are smaller, then at maximum capacity, wouldn't the mean lake depth also be smaller? Maybe I misunderstand the interpretation of ‘mean maximum lake depth’. Is this the mean depth of all water pixels? Or the average of the maximum lake depth for all lakes? Or maybe a change to the word “smaller” would be helpful.

We mean the mean depth of all water pixels. The previous changes to indicate that this is a spatial average clarify this point. Additionally, we have added the following modification:

“Lower Hurst exponents indicate smaller more frequent bumps along a rough surface and correlate with higher mean maximum area fraction values as shown at the Larsen C ice shelf (Figure 9).”

L380 – what, specifically, is meant by “melt lake characteristics”?

Melt Lake Characteristics would be referring to the melt lake depth. Lined has been changed to clarify.

Specific Comments

L12 – Change “ice sheet surfaces” to “the Antarctic Ice Sheet surface” since you only focus on Antarctica in this work.

In this study, we first use extensive surface elevation measurements from the ICESat-2 satellite altimetry mission to show that roughness on the **Antarctic Ice Sheet surface** is largely self-affine, consistent with prior observations of bed roughness beneath ice sheets and geomorphic surfaces more broadly.

L33 –“As melt causes ice sheet surface warming...” This sentence is a bit strangely worded. Consider changing to something like: “Ice sheet surface melting initiates a positive feedback loop, whereby a lower surface albedo allows for more shortwave absorption, further enhancing meltwater production and deepening lakes. ” Or something similar.

Modified to:

“Ice sheet surface melting initiates a positive feedback, whereby a lower surface albedo allows for more shortwave absorption, further enhancing meltwater production and deepening lakes.”

L34 – add “surface” before “albedo”

Modified to:

“Ice sheet surface melting initiates a positive feedback, whereby a lower **surface albedo** allows for more shortwave absorption, further enhancing meltwater production and deepening lakes.”

L35 – change “causes” to “promote”

Modified to:

“After refreezing, albedo is permanently altered from its original value due to snowpack and firn density changes from meltwater saturation, which **promotes** future melt lake formation at these locations...”

L47 – change “fracture” to “fracturing”

Modified to:

“Incorporating supraglacial melt lakes into large-scale models is thus vital in properly simulating the potential interaction between surface melt and ice sheet **fracturing** and calving.”

L84-87 – “The ATLO6 data product...” For readability, reword this sentence to: “The ATLO6 data product measures the land-ice elevation and is a processed version of the raw photon data which measures the travel time between the satellite and the Earth’s surface (available as the ATLO3 product, though not used in this study)”.

Modified to:

“The ATLO6 data product measures the land-ice elevation and is a processed version of the raw photon data which measures the travel time between the satellite and the Earth’s surface.”

L128 – For “little variation” (twice in this line), please specify temporal or spatial.

Modified to:

“... we found little **temporal variation** in surface roughness parameters over that period. Figure 2 also shows little **spatial variation** of the Hurst exponent across the ice sheet.”

L139 – could “bumps” be changed to something like “topographic variability”?

Modified to:

“This indicates the possibility that processes related to snow redistribution and densification may influence surface topography, which creates smaller-scale **topographical variability** [Lowe et al., 2007].”

L157 – Add “temporally” to “... assuming the ice sheet surface does not change”.

Modified to:

“Since water gathers in surface depressions, the statistics of roughness on the surface can be used to determine the statistics of lake size on that surface, assuming that the surface topography does not change **temporally**...”

L208 – Move (w) after “average melt lake depth” (L207) for clarity.

Modified to:

“For each surface, we calculate the average melt lake depth (w_i)...”

L211 – Change “for” to “to calculate”

Modified to:

“Additional work is done **to calculate** the mean water depth...”

L323 – specify temporal or spatial “variations”

Modified to:

“Additionally, we assume that topographical parameters measured by the ICESat-2 mission in 2021-2022 apply to predictions over the 2014-2018 period when observations are available due to the relatively small **temporal variations** in these parameters throughout the ICESat-2 mission.”

L403 – Change “Due” to “Given”

Modified to:

“**Given** to the relatively minor variation in Hurst exponents...”

L414 – Specify “Greenland subglacial topography”. Also, why might the Hurst exponent for subglacial topography differ from that for terrestrial geomorphic surfaces?

Fixed. This is a complicated question and one that we are hoping to address in ongoing work. It is likely due to the fact that subglacial topography is subject to qualitatively different erosive processes than surface ice sheet topography. However this is beyond the scope of this manuscript.

References

Lutz et al., (2024). Assessing supraglacial lake depth using ICESat-2, Sentinel-2, TanDEM-X, and in situ sonar measurements over Northeast and Southwest Greenland, The Cryosphere. <https://doi.org/10.5194/tc-18-5431-2024>.

We sincerely appreciate the constructive feedback provided by the reviewers. We have addressed all suggested edits. Below you can find reviewer comments in blue and our responses in black.

Reviewer #1 (Remarks to the Author):

Review of “Predicting Mean Depth and Area Fraction of Antarctic Supraglacial Melt Lakes with Physics-Based Parameterizations”

by Grau et al.

submitted for consideration in Nature Communications

Summary

The authors implemented relatively modest revisions in order to address comments made by two reviewers during the previous round of review. While many of the revisions improved the manuscript, I found that a few of them were not beneficial. Below, I call attention to the detrimental revisions and offered some recommendations for how they can be improved. A handful of other revisions are also included below.

Major Comments

1. You do not explicitly define the terms in Eqn. 2. For example, presumably h is the ICESat-2 elevation but that needs to be stated. Additionally, the bounds of the integration are listed as infinite but your sub-tracks are 10 km in length and it seems as though your frequencies are restricted to $10^{-3.5}$ - $10^{-9.2}$ (based on Fig. 2) but that I not defined or explained. Please provide more detail in the formulation of this equation.

The bounds of integration have been changed to [0 to 10 km], which covers the full range of length scales in a Icesat-2 subtrack. Additionally, we have defined variables throughout the text prior to the derivation of the Hurst exponent. Figure 2 has also been updated to display the power spectral density in a loglog plot with respect to the wavenumber. The slight curving towards the higher frequencies in this loglog plot has also been explained further in the figure caption:

“The slight deviation from the linear regression at higher wave numbers is due to the ATL06 product processing aggregate photon counts in 40 meters intervals (Brunt et al. 2019), thus artificially smoothing roughness at the highest resolutions of the product”

2. In my opinion, Figure 1 does not help convey self-affinity. I always imagine snowflakes when thinking about fractals (see https://personal.math.ubc.ca/~cass/courses/m308-03b/projects-03b/skinner/ex-dimension-koch_snowflake.htm). I like the multi-scale approach, with the zoom insets, but each zoom should show the same level of complexity but at different scales. I imagine something in which large chevron-shaped lakes are visible

at the largest scale and you zoom in once or twice to show that there are also smaller chevron-shaped lakes that are not as visible at the larger scales. That would be much more effective.

Figure 1 has been updated substantially to emulate the self-repeating pattern at different horizontal length scales referenced by the reviewer. We have drawn inspiration from prior illustration of self-affine geomorphic roughness in updating this figure.

3. For Figures 8 and 9, the color scheme does not appear correct based on the legend. Based on my interpretation of these figures, the darker blue color brackets the 25-50th percentiles and the lighter blue brackets the 5-95th percentiles. The legends suggest the opposite.

Figure legends have been corrected.

4. I appreciate the inclusion of two different predictions for the validation regions but I think that the discussion of why either the area or depth parameterization is in error is still lacking. One of the major discussion points that is missing is a discussion of the elevation data sub-track size and the frequency range used for solving the Hurst values. In Figure 2, you can see that the slope of the line is not the same across the full range of frequencies under consideration. If you focused on frequencies from $\sim 10^{-5}$ to $\sim 10^{-3}$ only, you would have a much shallower slope for that best-fit line. You mention the influence of the Hurst exponent on results in lines 382-388 but that discussion tries to argue that Antarctica has little variation in the exponent and does not mention how your interpretation of the data could potentially influence the agreement with observations.

We have done additional tests on the sensitivity of the predictions to the Hurst exponent and have found that changing the Hurst exponent ± 0.2 from the mean $H = 0.4$ doesn't not alter the predictions significantly due to the weak dependence of area fraction on Hurst exponent. We have added a sentence indicating this lack of sensitivity in Section 4.3. We note that in the example given, the main departure from a Hurst exponent of 0.4 is at very small horizontal length scales. Since most supraglacial melt ponds are hundreds to thousands of meters in horizontal dimension, small inaccuracies in the roughness at the smallest length scales do not play a significant role in setting melt lake size. We have also added a note in the caption of Figure 2 explaining that this departure at the highest frequencies of the PSD is a result of how the ATL06 product is processed.

Line-by-line Comments

- Lines 20-21: The new sentence added at the end of the abstract is somewhat awkward and does not accomplish the goal of articulating the impact of the work. Please revise further to circle back to the first two sentences that focus on albedo and ice shelf stability. Your penultimate sentence describes potential limits on pond size and depth. What are the potential implications related to albedo and hydrofracture?

We have modified the abstract as follows:

“In this paper, we present a simple method for parameterizing supraglacial melt lakes in large climate and ice-sheets model that has an intrinsic limit in potential future melt lake size. The implementation of these parameterizations will improve the simulation of albedo and hydrofracture of water-filled crevasses in large-scale climate and ice sheet models. Ultimately, such parameterizations can be used to improve predictions of future ice sheet and ice shelf collapse in response to climate change.”

- Lines 91-92: This sentence still reads as though you only select certain weak beams because you cannot expect the reader to know that sometimes data is not available from all three beams. Please rephrase to indicate you use all available weak beams.

Line has been changed to:

“We select the first available weak beam from each data file in order not to double count data from both tracks.”

- Lines 122-129: Why not include the same level of detail about filtering as you described in the response to the reviewers?

Paragraph has been altered to:

“All sub-tracks with calculated Hurst exponents less than zero or greater than one are discarded, since they cannot be reliably described as self-affine (i.e. outside the acceptable confines of the range for the Hurst exponent). Typically, such cases are related to spurious elevation data causing large deviations in the power spectral density. Additionally, any linear regressions with R^2 values less than 0.7 are discarded to ensure that the only sub-tracks considered are those in which the data strongly indicate roughness is self-affine. Of the 1.8 million sub-tracks that were analyzed, 7.3% of the sub-tracks were discarded during the analysis process. 7.1% were discarded due to poor R^2 values when fitting, and the remaining 0.2% were discarded due to the calculated Hurst exponents being outside the acceptable range of [0,1]. As a result, over 92% of all the subtracks analyzed exhibit the qualities of a self-affine surface. Consequentially, we conclude that the Antarctic Ice Sheet surface is largely self-affine.”

- Line 203: You do not allow percolation into the subsurface so I would not use the term “percolation” to describe movement of water across the surface.

“Percolation” has been changed to “Movement”.

- Line 231-232: I agree that you need to explicitly state that you do not permit percolation into the subsurface but this statement should come in the previous paragraph.

We are not certain whether this comment applies to these lines, but we have moved the statement to indicate that the conditioned walker model does not simulate subsurface percolation earlier in the description of the model. Line 209-210 now reads:

“This is under the assumption that the surface is non-porous with none of the precipitons penetrating the subsurface.”

- Line 306 and onwards: Why is the “s” in Landsat capitalized here when it is typically not in literature?

All of the “s”s in Landsat have been uncapitalized.

- Lines 307-318: I like the additional detail that was added but this paragraph could be condensed. Sentences 2-4 essentially repeat the same points but with slightly different details.

The paragraph has been reduced to four sentences. Some details have been kept due to other reviewer’s concern for understanding how the validation data set it is computing melt lake depths.

- Lines 336-337: Please rephrase. The revisions to this sentence did not improve readability.

Line had been rephrased to the following:

“This results in a distribution of monthly predictions calculated from equations 11 and 12, with inputs taken over the area where supraglacial lakes are observed in Landsat, and including: surface roughness parameters estimated as described in Section 2, and surface melt supply from RACMO.”

- Lines 416-417: I would start this paragraph with “Our melt lake parameterization suggests...” or something along those lines. Currently it is unclear if you are making a statement about albedo based on your work or you should be referencing literature.

Line has been altered to the following:

“Our melt lake parameterization suggests that the albedo effect of melt lakes is not any greater on rougher ice surfaces than on smoother ones, since the area fraction of melt lakes does not depend on the surface roughness amplitude.”

- Line 422: Broken reference.

Citation has been fixed in Line 422.

- Line 477: That GitHub repository link is broken.

Reviewer #1 (Remarks on code availability):

The link took me to a 404-error page.

Link has been adjusted and should take user to the corresponding repository.

Reviewer #2 (Remarks to the Author):

I appreciate the work that the authors put into the second version of this manuscript. The methodology is much clearer and I appreciate the addition of Figure 5. I agree with all of the changes made and only have a few minor additional comments/remarks:

Figure 1 – I'm not really sure what I am supposed to get out of the Figure. I think some labels and a more descriptive caption would be helpful. What are the blue spots? Also, I don't think this Figure is referenced in the text (please check this for other figures as well)

We have updated Figure 1 substantially to demonstrate an idealized self-affine glacier inspired by Figure 1 in Cael et al., 2017. The blue spots are supraglacial melt lakes on the glacier. We have added more detailed explanation in the caption.

In Figure 5 – does the height of the cells in the cellular automaton model indicate their relative elevation? It might be helpful to label the 'z' axis in this figure.

The height of the of the cells in the cellular automaton model indicates their elevation. Axes labels have been added to this figure.

L272 – "The sentence beginning with "As the depth of the melt lake..." does not make sense

Sentence has been rewritten to:

"On a self-affine surface, the average horizontal extent of lakes filling depressions does not depend on the average roughness height (Cael et al. 2017). Therefore, we expect that the area fraction of supraglacial melt lakes should have no dependence on σ ."

L317 – Include space after equation

Space was added after equation.

Figures, in general – Many of the figures have random horizontal and vertical lines around parts of the subfigures (eg. Figure 3, 11). This could be just in the rendering of the PDF but perhaps check to make the figures cleaner.

We have determined this is a rendering issue in Adobe Acrobat. We have remade the figures to eliminate the issue.

Review of “Predicting Mean Depth and Area Fraction of Antarctic Supraglacial Melt Lakes with Physics-Based Parameterizations”

Grau et al. 2024

Summary

In this manuscript, the authors provide parameterizations for supraglacial melt lake characteristics that could be used in future ice sheet modeling work. Overall, this is an interesting manuscript. It is well-written and the figures are generally clear. I have some major concerns relating to the methodology. Once these concerns are addressed, I believe this work will be a good contribution as parameterizations for supraglacial meltwater will be increasingly important in ice sheet modeling.

One general comment: The structure of this manuscript is a bit odd. Sections 2-4 each have individual methods and results subsections. I don't necessarily mind this structure and I found it relatively intuitive and easy to follow, but I'm not sure that it aligns with the journal requirements. If the current structure is maintained, I would recommend using more verbose titles for sections 2.2, 2.3, 3.1, 3.2 and 4.3 so there are not multiple subsections titled simply “methods” or “results”. For example, Section 2.2 could be renamed “AIS self-affinity methodology” or something of the sort.

Major comments

Generally, I find some methodological details lacking. I also believe that an overarching diagram or schematic of all the methods would be helpful.

Self-affinity (Section 2)

- what quality proves that a surface is self-affine? It is unclear to me from this work, what actually demonstrates that the AIS is self-affine. Is it the Hurst exponent that proves self-affinity?
- The paragraph from L119-123 is confusing. In this section, I thought you were trying to demonstrate that the AIS is self-affine. But in these lines, the manuscript mentions that tracks are discarded if they are not described as self-affine or that “the only sub-tracks considered are those in which the data strongly indicate roughness is self-affine”. How can you demonstrate self-affinity if you self-select only tracks that are self-affine? Additionally, I find some details to be missing. For example: What percentage of the sub-tracks are selected? How many have $R^2 < 0.7$?

Mean lake depth and area fraction (Section 3)

- L206 – “with varying water supply depth, Hurst exponent, and roughness amplitude”. What do these values vary between? I guess this is in L232-233 but this information should come in the methods section. Also, more details are necessary. Are they randomly selected from the distributions found in Section 2 (i.e. Fig 2,3)? Can you

elaborate on the “water supply depth”? Is this the same as the “meltwater availability”? Where does this information come from?

- L232 – is it realistic to do this over the range of possible H values [0,1] since it seems like >99% of the calculated H values fall between 0.3 and 0.6 (Fig. 2).

Validation of lake parameterizations (Section 4)

- Can you provide the equation used to calculate depth? Which bands are used? What are the limitations and uncertainties associated with these depth calculations? Throughout this section it seems that you treat the ‘observed depths’ as the truth. However, there are large uncertainties when using optical imagery to determine lake depth. A greater discussion of this is necessary.
- It appears from Figure 6 that for the lake depth and area fraction predictions, it is assumed that meltwater refreezes before July? In reality, melt refreezes much before then. Can you better temporally constrain the predictions? Maybe using RACMO air temperature?
- Is there a way to include firn air content in this parameterization, thereby accounting for percolation?
- L344 – what is the scaling factor used?

Discussion (Section 6)

- L369 – “The parameterizations developed in this study are mostly consistent with observations.” I’m not really convinced by this. I am especially concerned with Figure 6b/d. Considering there are huge uncertainties in both the observations (lake depth from optical imagery) and the parameterizations (not accounting for firn percolation), I’m not convinced by the comparison to observations.
- L390 – which areas specifically are most at risk of future lake expansion? Could there be some more advanced future insight included in this manuscript, besides just that other areas may be at risk of melt pond development?
- L392-393- “This may explain why, ... observed lake depths span a relatively narrow range”. Or perhaps, the algorithm used to calculate lake depth has saturated. Previous work (e.g. Lutz et al 2024) has shown that lake depths computed using the red band saturate at low depth, therefore a full range of lake depths cannot be calculated using this method. This limitation should really be discussed in the comparison to observations.

Minor comments

Abstract – I think you need a concluding sentence in the abstract that summarizes the motivation and importance of this work. This should tie back to the first 2 sentences of the abstract. As written now, the significance of this work is not immediately clear from the abstract.

L21 – I don't think that the Kingslake et al. (2017) paper showed an increase in surface melting and supraglacial lake formation.

L31-43 – This paragraph goes into a lot of detail about the impact of supraglacial meltwater on surface energy balance and energy fluxes. This seems like too much detail considering the manuscript is not about ice sheet surface energy balance. I would appreciate more background introduction on topography theory instead of energy fluxes.

L48 – “Supraglacial melt lakes are omitted from *most* current large-scale models...” Are there any models that incorporate them?

L53 – The last sentence of this paragraph should be reworded to be more specific and stronger regarding the contribution of this work.

L55-59 – It would be helpful to have more background information here

L59-69 – Relate the work done back to the original knowledge gaps.

L111 – “... provides similar results to other methods.” Did you compare to other methods, or was this found in a previous study?

L130 – “There is a clear difference in Hurst exponent between grounded and floating ice”. Is this difference statistically significant?

L149 – “It was found that the Hurst values do not vary much...” How much? Can this be quantified in the main text?

Figure 2,3 – Why do you show a PDF for the Hurst exponent and a CDF for the standard deviation?

L167 – can you provide a brief description/explanation of “cellular automata”?

L174 – “meltwater runoff depth” This is the same as “meltwater availability”, right?

Section 3.1 – Somewhere in this section it is important to mention that these methods assume a non-porous surface and that there is no infiltration and all precipitons pond on the surface.

L216 – Can you elaborate what is meant by “custom equation forms”?

L234 – when writing the “ $H - \sigma$ pair” (and later for $H - \sigma - w_s$), is there a better way to write this that does not look like subtraction? Maybe (H, σ)

L245 – “... we create a simple mathematical expression...” How is this created? Can more detail be provided?

Figure 4/5 – Please ensure that all color bars are labeled. Also, in 4a, for large σ , why is the average lake depth higher for lower total water supply? In Figure 5, the relative difference is almost always negative. This may indicate that F is not parameterize properly?

L309 – Specify the “predictions” you aim to provide (e.g. mean lake depth, lake area fraction).

L314 – What is meant by “the regions within the two ice shelves where melt lakes are observed”?

Figure 6 – what do the different shades of blue represent? Why is it often cut-off at the top of the plot? The y-axis should be extended to show the full range of predicted values. Also, would it be possible to include RACMO meltwater production on these plots?

L361 – When thinking about maximum lake depth, it does not make sense to me that “smaller more frequent bumps” would “correlate with a higher mean maximum water depth”. If the bumps are smaller, then at maximum capacity, wouldn’t the mean lake depth also be smaller? Maybe I misunderstand the interpretation of ‘mean maximum lake depth’. Is this the mean depth of all water pixels? Or the average of the maximum lake depth for all lakes? Or maybe a change to the word “smaller” would be helpful.

L380 – what, specifically, is meant by “melt lake characteristics”?

Specific comments

L12 – Change “ice sheet surfaces” to “the Antarctic Ice Sheet surface” since you only focus on Antarctica in this work.

L33 – “As melt causes ice sheet surface warming...” This sentence is a bit strangely worded. Consider changing to something like: “Ice sheet surface melting initiates a positive feedback loop, whereby a lower surface albedo allows for more shortwave absorption, further enhancing meltwater production and deepening lakes.” Or something similar.

L34 – add “surface” before “albedo”

L35 – change “causes” to “promote”

L47 – change “fracture” to “fracturing”

L84-87 – “The ATLO6 data product...” For readability, reword this sentence to: “The ATLO6 data product measures the land-ice elevation and is a processed version of the raw photon

data which measures the travel time between the satellite and the Earth's surface (available as the ATLO3 product, though not used in this study)".

L128 – For “little variation” (twice in this line), please specify temporal or spatial.

L139 – could “bumps” be changed to something like “topographic variability”?

L157 – Add “temporally” to “... assuming the ice sheet surface does not change”.

L208 – Move (\bar{w}_l) after “average melt lake depth” (L207) for clarity.

L211 – Change “for” to “to calculate”

L323 – specify temporal or spatial "variations"

L403 – Change “Due” to “Given”

L414 – Specify “*Greenland* subglacial topography”. Also, why might the Hurst exponent for subglacial topography differ from that for terrestrial geomorphic surfaces?

References

Lutz et al., (2024). Assessing supraglacial lake depth using ICESat-2, Sentinel-2, TanDEM-X, and in situ sonar measurements over Northeast and Southwest Greenland, *The Cryosphere*. <https://doi.org/10.5194/tc-18-5431-2024>.